# Quantum coordinates, localisation of events, and the quantum hole argument

Viktoria Kabel [1,2,5] ✉, Anne-Catherine de la Hamette [1,2,5], Luca Apadula[1,2,5], Carlo Cepollaro [1,2],
Henrique Gomes[3], Jeremy Butterfield [4] & Časlav Brukner[1,2]

The study of quantum reference frames (QRFs) is motivated by the idea of taking into account the quantum properties of the reference frames used, explicitly or implicitly, in our description of physical systems. Like classical reference frames, QRFs can be used to define physical quantities relationally. Unlike their classical analogue, they relativise the notions of superposition and entanglement. Here, we explain this feature by examining how configurations or locations are identified across different branches in superposition. We show that, in the presence of symmetries, whether a system is in "the same" or "different" configurations across the branches depends on the choice of QRF. Hence, sameness and difference — and thus superposition and entanglement — lose their absolute meaning. We apply these ideas to the context of semi-classical spacetimes in superposition and use coincidences of four scalar fields to construct a comparison map between spacetime points in the different branches. This reveals that the localisation of an event is frame-dependent. We discuss the implications for indefinite causal order and the locality of interaction and conclude with a generalisation of Einstein's hole argument to the quantum context.

When describing a physical system, it is very common to do so with respect to a reference frame—a ruler used to determine the position of a particle, for example, or a clock, which tracks the time that elapses while it is moving. Furthermore, one can take the view that "the particle's position" or "the tracking of time" are meaningful only in relation to the degrees of freedom of the system that serves as the reference frame or clock. Such a relational account of physical systems and dynamics seems to be a fundamental aspect of how we describe the world. Usually, the reference systems that we employ, either explicitly or implicitly, are treated as purely classical objects with well-defined properties. But what happens if we take into account the possibility that the reference system itself exhibits quantum properties? This question has motivated a recent wave of research on *quantum reference frames* (QRFs)[1–23], investigating what the world looks like with respect to a quantum system. In particular, it has been found that the quantum properties of the frame, such as superposition or entanglement, get transferred onto the relational description of the surrounding systems. As a result, a QRF transformation can change whether a system appears to be in superposition of different configurations, and whether it is entangled with another system, thus rendering these features frame-dependent[2,8]. If we assume that the covariance of physical laws under classical changes of reference frame extends linearly to encompass changes of QRF[2], then the latter should not affect the physical situation. Thus, whether a system is in a superposed or entangled state, and how these properties change dynamically, becomes a mere matter of perspective.

The frame-dependence of quantum properties is already well understood at the formal level, within several different frameworks for QRFs[1–3,7,8,13,14,22–24]. However, the meaning and usage of QRFs is still not fully clarified on a conceptual level. The purpose of the present work is to abstract from the mathematical details of a subset of the various formalisms and applications and to crystallise their common core by embedding several recent works on QRFs[2,5,8,11,17–20] into a single unified picture. It provides a visualisation of different choices of QRFs and how they are connected. This, we believe, can greatly enhance the intuitive understanding of QRFs and allow for a more pedagogical introduction into the subject. With the help of concrete toy examples, we offer a new understanding of the frame-dependence of superposition and entanglement under QRF changes: Ultimately, whether a system is in "the same" or "different" configurations across the branches—and thereby, whether it is in a state of superposition or not—depends on which QRF we use to identify and compare these configurations. These notions of identification and comparison change under a QRF transformation. As a result, sameness and difference lose their absoluteness.

[1]Institute for Quantum Optics and Quantum Information (IQOQI), Austrian Academy of Sciences, Boltzmanngasse 3, A-1090 Vienna, Austria. [2]University of Vienna, Faculty of Physics, Vienna Doctoral School in Physics and Vienna Center for Quantum Science and Technology (VCQ), Boltzmanngasse 5, A-1090 Vienna, Austria. [3]Oriel College, University of Oxford, OX14EW Oxford, UK. [4]Trinity College, University of Cambridge, CB21TQ Cambridge, UK. [5]These authors contributed equally: Viktoria Kabel, Anne-Catherine de la Hamette, Luca Apadula. ✉e-mail: viktoria.kabel@gmail.com

Making this statement precise may seem like a daunting task but fortunately, we can draw upon a wide range of existing work in the philosophy literature. Specifically, some of us[25,26] have recently constructed concrete comparison maps between different possible configurations—albeit on the classical level. The philosophical motivation for these comparison maps was the proposal by David Lewis[27,28] (pp. 38–43, and [29], Ch. 4) of what he called *counterpart theory*. For our purposes, his key ideas are that: (i) for objects to be the "same" in different physical situations (in philosophical jargon: different possible worlds) is for them to be similar in relevant respects or properties and (ii) what respects or properties count as relevant is "up to us", and so can change from one context to another (for more details, compare e.g. Section 2.2.3 of ref. [30]). By combining these insights with the "quantum" of quantum reference frames, we obtain a framework that allows us to understand more clearly how the identification between systems depends on the quantum frame. More concretely, we will discuss how the central components of the QRF formalism (as formulated in refs. [2,4,8,17] and parts of ref. [14]) and their applications[5,11,16,18–20] can be neatly embedded in the "space of models" developed in refs. [25,26,31,32] and how a choice of QRF picks out a particular comparison map. It is this change of the comparison map that is at the heart of many of the surprising consequences of QRF changes and which, to our knowledge, has not been spelled out in the QRF literature so far.

In addition to providing more intuitive access to QRFs from a purely quantum mechanical perspective, the conceptual nature of the framework also allows for its application to general relativistic scenarios. This enables us to explore the conceptual implications of invariance under QRF transformations for the diffeomorphism group in the context of semi-classical spacetimes in superposition (as considered, e.g., in refs. [16–19,33–38]). Note that when referring to "(semi-classical) spacetimes in superposition", we mean that the metric is in a superposition of states, each peaked around a classical configuration of the metric. More generally, in a slight abuse of language, we will often refer to a system/spacetime/manifold/etc. "in superposition"—this is a short form for the quantum state of the system/spacetime/manifold/etc. being in superposition. As was stressed by Hardy in his work on the quantum equivalence principle[39], this regime is of particular interest as it might allow us to better understand spacetime at the quantum level, similar to how the classical diffeomorphism invariance of general relativity has significantly illuminated our classical understanding of spacetime. Note that we do not want to claim that we solve the many mathematical difficulties in dealing with the diffeomorphism group and general spacetime metrics, in particular the problems of defining well-behaved measures. Rather, we focus on what we *can* say by making a few core assumptions while these problems remain open. We find that, in this context, a choice of QRF comes with a notion of how to identify spacetime points across the superposition. Note that a priori there does not exist any preferred way of comparing or identifying the locations of objects across different spacetimes in superposition. There have been some attempts in the literature at constructing such a "threading" of points between the different manifolds, on the classical[40] and the quantum level[39,41]. In refs. [33,34], the authors take a first step towards implementing such threading in the context of QRFs by using the location of a probe particle to identify some points of the different manifolds in superposition. Nevertheless, in a diffeomorphism-invariant theory, the position of a single particle is not sufficient to specify how to identify points across entire spacetimes in superposition. Here, we extend this idea by replacing the probe particle by four scalar fields and using their coincidences to construct a comparison map between all spacetime points in the superposition. This provides us with the necessary structure to define whether an event is located at "the same" or "different" points across the spacetimes in the superposition—whether it is localised or delocalised. While we show that the localisation of events depends on the choice of QRF, we argue that, nonetheless, QRF transformations do not have empirical consequences for phenomena such as interference and recombination. Moreover, the insight that localisation is frame-dependent helps to shed light on the notion of a "quantum event" and in particular the question of how to count events in the context of superpositions of spacetimes, which

also plays an important role in the study of indefinite causal order[19,42–47]. Furthermore, the understanding of QRFs as providing a preferred identification of points across spacetimes in superposition has interesting implications for a possible quantum generalisation of the hole argument, which we discuss at the end of this article.

## Results and discussion
### The space of models and representational conventions

One important goal of our work is to provide a unified framework for QRFs, applying to a variety of different theories with varying symmetry groups. In this section, we recapitulate the ideas of refs. [25,26,30–32] on symmetries and "representational conventions" and explain how QRFs fit neatly into this framework once one makes a few conceptual extensions so as to adapt the ideas to the quantum level.

Consider a theory with symmetry group $G$. At this stage, we need not place any restrictions on the type of theory, besides the assumption that its symmetry can be represented by some group $G$. (For a precise definition of what constitutes a symmetry for a general theory, see pp. 3–4 of ref. [31].) Denote the state space of the theory by $\Phi$. A point in the state space will be given by a particular configuration or model $\varphi \in \Phi$. Following refs. [25,26,31,32], we will often refer to $\Phi$ as the *space of models*. As an example, consider general relativity. In this case, a particular model $\varphi = (\mathcal{M}, g, \Psi_{\text{matter}})$ consists of a manifold $\mathcal{M}$, a (Lorentzian) metric $g$, and, in general, some matter fields $\Psi_{\text{matter}}$. The symmetry group, in this context, is the diffeomorphism group $\text{Diff}(\mathcal{M})$. Now, given a symmetry group $G$, the space of models is partitioned into orbits of $G$. Along a single orbit, the models are mapped into each other via the action of the group (see Fig. 1). More concretely, denoting by $\varphi^g$ the result of acting on $\varphi \in \Phi$ with $g \in G$, the orbit of $\varphi$ is defined as $\mathcal{O}_\varphi = \{\varphi^g, g \in G\}$.

In the case of general relativity, one orbit would contain all diffeomorphically equivalent spacetimes; that is, including the metric and matter fields. Because $G$ is a symmetry group of the theory, we say that models in the same orbit are *physically equivalent* or *indistinguishable*. Physically distinct models, such as $\varphi$ and $\varphi'$, are in different orbits. As a consequence of the symmetry of our theory, the space of models contains a lot of redundancy—to obtain a complete description of all physically distinct states of the system our theory is describing, it would suffice to know one representative of each orbit. In gauge theories, one usually deals with this redundancy by fixing a gauge. In generally covariant theories, one fixes local coordinate systems. What is often implicitly assumed in these procedures is that there is a unique prescription, such as the Coulomb gauge condition $\vec{\nabla} \cdot \vec{A} = 0$ in electrodynamics, which applies to all the models equally. In the following, we are mainly going to be referring to gauge symmetries and gauge fixings rather than general covariance and coordinate choices. However, our framework applies to generally covariant theories as well, as we will see later. That is, the same gauge condition is imposed on all the different orbits $\mathcal{O}_\varphi$ to pick out a

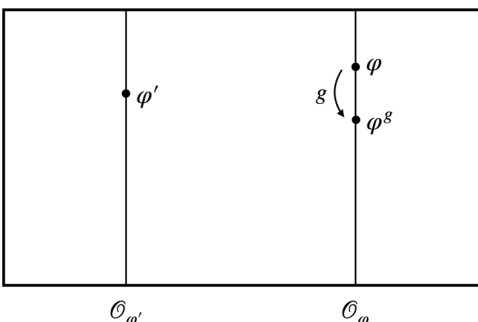

**Fig. 1 | The space of models $\Phi$.** Each point corresponds to a model $\varphi$—a particular configuration of the system under consideration. A given symmetry group $G$ partitions the space of models into orbits $\mathcal{O}_\varphi$ of models that are related to one another by a group action, such as $\varphi$ and $\varphi^g$. Models belonging to the same orbit are taken to be physically equivalent; models on different orbits, such as $\varphi$ and $\varphi'$, to be physically distinct.

preferred representative $\varphi^* \in \mathcal{O}_\varphi$. This, however, is not necessary; indeed, in theories more complicated than electrodynamics, it is not even possible. The Gribov obstruction[48] shows that no single such prescription can cover all of the orbits (note that it has also been explored in the context of QRFs in refs. 4,49,50). The symmetry of the theory grants a much larger freedom in the choice of representatives. In principle, we can let the way we choose a representative vary arbitrarily between the different orbits. In other words, we can use *different* gauge fixing prescriptions for different possible states of the world. This crucial insight also lies behind the QRF formalism, but we will discuss this in more detail below. The most general way of removing the redundancy in our description is to simply pick one unique representative for each orbit. This can be achieved via an injective map

$$\sigma : [\Phi] \to \Phi$$
$$[\varphi] \mapsto \sigma([\varphi]) \qquad (1)$$

where $[\Phi]$ denotes the space of equivalence classes $[\varphi]$ with respect to the equivalence relation "~", which is defined by $\varphi \sim \varphi'$ if and only if there exists a $g$ such that $\varphi' = \varphi^g$. In ref. 32, the idea of this injective map $\sigma$—which, in general, does not necessarily respect all structures on $\Phi$, such as being differentiable—is called a *representational convention*. But of course, one can require such structures to be preserved. For gauge theories, with a space of models which has a principal fibre bundle structure, $\sigma$ would be a (not in general global) section or slice (see e.g. refs. 51,52) through the fibre bundle.

Figure 2 illustrates two particular choices of representational convention $\sigma$ and $\tilde{\sigma}$ in the space of models. In practice, it can be useful to map between these different descriptions. Such a map can be implemented by applying a *different* group element in different orbits. More concretely, for each orbit $\mathcal{O}_\varphi$, we can find a unique group element $g_{\sigma \to \tilde{\sigma}}^{(\mathcal{O}_\varphi)}$, which takes $\sigma(\varphi)$ to $\tilde{\sigma}(\varphi)$ (see Fig. 2). In general, these group elements will differ across the orbits. A map between two different representational conventions can thus be seen as acting with an orbit-dependent choice of group elements.

Within the space of models, symmetry can now be understood as freedom in the choice of representational convention. In other words, a *change* of representational convention does not change the physical content of our description.

Finally, given a representational convention, one can define a *counterpart relation* which allows one to compare different models, that are in general in different orbits and thus represent different physical states. Or in philosophical jargon, the counterpart relation allows one to compare different possible worlds. In the presence of symmetries, this comparison is non-trivial. Thus in general relativity, one needs to ask: What do we mean by "the same" or a "different" point in spacetime when considering different solutions to Einstein's equations? Similarly in electromagnetism: what is "the same" or a "different" value of the gauge potential across various

possible configurations of the electromagnetic field? Usually, we answer these questions by fixing a coordinate system or gauge across all the different configurations. More generally, one can do so by fixing a representational convention. Given a representational convention $\sigma$, and two models $\varphi$ and $\varphi'$, we define a counterpart relation, written $\text{Counter}_\sigma(\varphi, \varphi')$, that is itself an element of the symmetry group $G$. Roughly speaking, it is the group element that, combined with "horizontal movement" across the section, transforms $\varphi$ to $\varphi'$. More precisely, we define, following[25,26]

$$\text{Counter}_\sigma(\varphi, \varphi') = g_\sigma(\varphi')^{-1} g_\sigma(\varphi) \qquad (2)$$

which prescribes how to compare two models $\varphi$ and $\varphi'$. Here, $g_\sigma(\varphi)$ is the group element that takes $\varphi$ to the surface defined by the representational convention. The counterpart relation thus tells us how to compare two models $\varphi$ and $\varphi'$: by taking $\varphi$ "down" to the section $\sigma$ using $g_\sigma(\varphi)$ and then raising it again using $g_\sigma(\varphi')^{-1}$, $\varphi$ has the same "alignment" as $\varphi'$ with respect to $\sigma$. This is illustrated in Fig. 3. Note, however, that we are using a (trivially) different group action than[25,26]. While they implement the (left) group action by acting with the inverse group element from the right, we act with the original group element on the left. To get a better feeling for the counterpart relation, let us see how models are compared in specific examples:

- If the states $\varphi$ and $\varphi'$ already lie on the same section or have the same alignment with respect to it in the sense that $g_\sigma(\varphi) = g_\sigma(\varphi')$, then the counterpart relation is the identity: $\text{Counter}_\sigma(\varphi', \varphi) = \text{Id}$. That is, within a given representational convention, we can use the identity element of $G$ to compare $\varphi$ and $\varphi'$.
- If the two states lie on the same orbit, the counterpart relation will simply be the group element relating the two, no matter the choice of section: $\text{Counter}_\sigma(\varphi, \varphi^g) = g$. That is, within a single orbit, the group element that encodes the comparison of models is section-independent.

Note that the counterpart relation does not tell us *simpliciter* whether two models are the same—nor whether they are physically equivalent, that is, lie in the same orbit. Rather, the counterpart relation gives a *meaning* (in fact, a section-dependent meaning) to the assertion that they are the same. In particular, take two models $\varphi$ and $\varphi^g$ on the same orbit. Then we should assert that these are physically the same even though they are not identical as models because we ought to compare them using the appropriate counterpart relation—the group element $g$—rather than the identity Id, independently of the section. More generally, any two models in $\Phi$ can either be identical or different. If they are different, their difference may be attributed to a redundancy in the description or it may not. If it is, a symmetry transformation will map the two models onto one another and the counterpart relation will take this into account; if it is not, no such transformation exists.

While finding a means of comparison for different models, i.e. different possible configurations of the system of interest, may seem like a purely

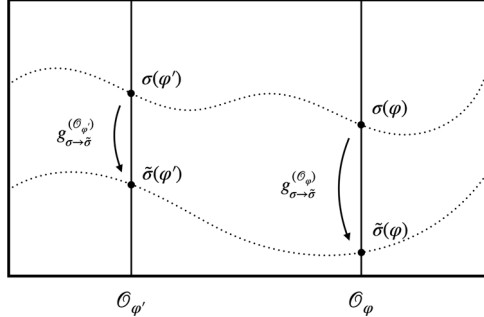

**Fig. 2 | A change between two representational conventions $\sigma$ and $\tilde{\sigma}$.** A representational convention $\sigma$ picks out one unique representative $\sigma(\varphi)$ from each orbit $\mathcal{O}_\varphi$. The upper dotted line shows the section through the space of models defined by $\sigma$ while the lower line corresponds to $\tilde{\sigma}$. For each model $\sigma(\varphi)$ picked out by $\sigma$ (on the upper section), there is a *unique* group element $g_{\sigma \to \tilde{\sigma}}^{(\mathcal{O}_\varphi)}$, which takes it to the physically equivalent model $\tilde{\sigma}(\varphi)$ picked as a representative by $\tilde{\sigma}$ (on the lower section).

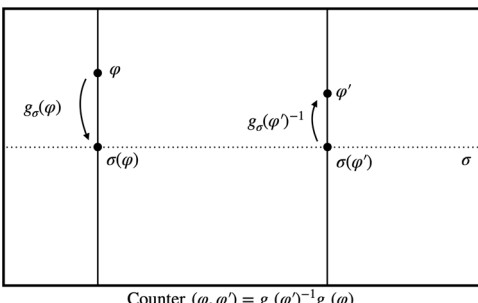

$$\text{Counter}_\sigma(\varphi, \varphi') = g_\sigma(\varphi)^{-1} g_\sigma(\varphi)$$

**Fig. 3 | The counterpart relation.** Given a representational convention $\sigma$, we can define the corresponding counterpart relation $\text{Counter}_\sigma(\varphi, \varphi')$ that prescribes how to compare models $\varphi$ and $\varphi'$. In particular, it aligns the two models by lowering $\varphi$ to the section $\sigma$ using $g_\sigma(\varphi)$ and then raising it again with $g_\sigma(\varphi')^{-1}$. In other words, it puts the two models on equal footing, allowing a direct comparison between different possible configurations.

philosophical question in the classical context, it becomes important even for the practicing physicist at the quantum level.

In order to see this and to make the connection to QRFs alluded to above, let us now make the crucial step that takes the notions introduced so far to the quantum level: rather than considering a set of models in $\Phi$ as a collection of *possible* configurations (states of our system), we take these models to denote the different branches of a quantum state of the system, i.e. branches or summands of a superposition. Note that, in this way, the symmetry group $G$ induces a preferred basis for models, in terms of which superpositions of configurations are expressed. The physical requirement, so as to distinguish such a superposition from a classical mixture, is to measure a quantity whose eigenbasis does not commute with the preferred one. In the present paper, we do not explicitly model such other bases. But of course, one can express states and observables on an arbitrary basis of the Hilbert space (setting aside subtleties that may arise in the context of field-theoretic models and in particular in the context of spacetimes in super-position that will be discussed below). In particular, in the translation group example described in the next section, the preferred basis is the position basis but one could straightforwardly express the states and observables in other bases as well. In general, it might be challenging to perform a mea-surement on a conjugate basis, in particular in the context of models involving spacetime.

Keeping these considerations in mind, let us now take the system's total state to be in a superposition of models $\varphi$ and $\varphi'$ (see Fig. 1) rather than treating $\varphi$ and $\varphi'$ as different yet unrelated possible states of our system. Importantly, we do not grant any physical meaning to superpositions of models that live on the same orbit and instead view them as equivalent to any single representative of the equivalence class. The orbit-wise freedom in the choice of the representational convention then translates to a *branch-wise fixing of the gauge* in the superposition of configurations. Note that in the following we will assume that the gauge-fixing section is chosen to be smooth across orbits. Moreover, a change from one representational con-vention to another is now implemented by a branch-dependent choice of which group element to act with—just like a QRF transformation (the idea of gauge transforming "each individual component of the superposition independently"[53],p. 2 also underlies the *complete quantum gauge transfor-mations* of Rovelli[53]). This strongly suggests the connection between representational conventions for configurations in superposition and QRFs: A specific choice of representational convention $\sigma$ corresponds to the choice of a QRF relative to which the state of the system is described. This is illustrated in several concrete examples further below.

The relation between QRFs and representational conventions not only equips us with a useful tool to visualise changes in QRF but also gives support to the generalised symmetry principles associated with these transformations. These symmetry principles have, so far, been taken to extend their classical counterparts by going from invariance under branch-independent symmetry transformations, to invariance under branch-dependent ones. In a sense, however, this extended symmetry is already present at the classical level in the free choice of representational convention. It is at the quantum level, however, that this becomes particularly relevant:

for it implies the frame-dependence of properties such as superposition and entanglement.

## The translation group: massive object in superposition

For the following example, consider a one-dimensional translation-invar-iant theory, i.e. choose the symmetry group $G$ above to be the one-dimensional translation group. This could, for example, refer to a set of $N$ point particles with an interaction that only depends on the relative distance as in refs. 2–4, or a simple gravitational system as in ref. 17. The space of all models of this theory can be partitioned into orbits, where each orbit con-tains configurations that can be related to one another by a rigid translation in position. A choice of representational convention corresponds to a choice of the origin for each model or, in other words, a choice of reference frame, which fixes the zero-point of the position. This picks out one unique representative of each orbit and thus (if done suitably smoothly) defines a section through the space of models.

Now consider a superposition of different models. A change from one representational convention $\sigma$ in the space of models to another $\tilde{\sigma}$ can now be seen as a change from one position reference frame (one stipulation of which point is the spatial origin) to another. A model-independent trans-formation, such as a standard classical reference frame transformation, would shift the origin by the same amount in each branch of the super-position. That corresponds to applying the same group element of the translation group to each model in the total configuration (see Fig. 4a). A model-dependent transformation, such as a *quantum* reference frame transformation, on the other hand, acts with a *different* group element, i.e. translation, in each branch (see Fig. 4b). This is commonly referred to as a *quantum-controlled* transformation.

In ref. 2, this is realised by associating the reference frame with a particular quantum particle. We thus choose the quantum particle as the quantum reference system and will from now on use the terms "reference frame" and "reference system" interchangeably, either expression merely emphasising different aspects of the same idea. More concretely, the zero-point of the coordinate system is taken to be the position of the chosen particle. A change between different QRFs can then be realised as follows: (i) "read off" the position $a$ of the new reference frame with respect to the old one (this distance will, in general, differ across the branches) and (ii) apply, in each branch, a translation $\mathbb{T}_{-a}$ by the established amount $-a$. For technical reasons, step (ii) is implemented in ref. 2 by (ii.a) translating all particles, apart from those that define the origins (zero-points) of the old and new reference frames, by the established amount $-a$; and (ii.b) exchanging the positions of the two (so far unmoved) particles that define the origins (zero-points) of the old and new reference frames, such that the new reference frame's particle is now at the origin, while the old reference frame's particle is at position $-a$. Represented in the space of models, this means that the translation $a(\varphi)$ can, in general, be different for each orbit $\mathcal{O}_\varphi$. This is already accounted for by the framework of representational conventions: the group element $g^{(\mathcal{O}_\varphi)}_{\sigma \to \tilde{\sigma}}$, which takes a model in the orbit $\mathcal{O}_\varphi$ on one section $\sigma$ to another section $\tilde{\sigma}$, depends on the orbit $\mathcal{O}_\varphi$.

As will be illustrated throughout the remainder of this section, a QRF transformation thus amounts to performing a different translation for each

**Fig. 4 | Graphical illustration of the difference between model-independent and model-dependent translations. a** Classical translations are commonly described as model-independent trans-formations, which shift the models by a constant amount by applying the *same* translation $\mathbb{T}_a$ in each branch of the superposition. **b** Quantum transla-tions, on the other hand, will, in general, apply *dif-ferent* translations $\mathbb{T}_a$ and $\mathbb{T}_{a'}$ in the different branches. As before, the dotted lines depict two sections associated with different representational conventions.

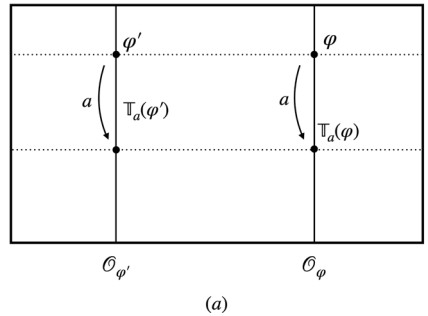
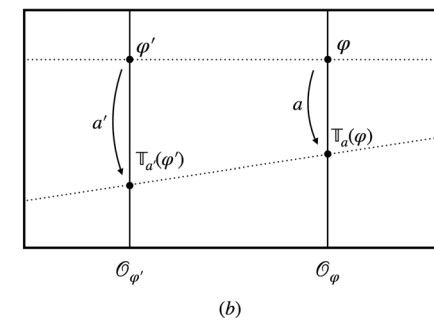

$(a)$ $(b)$

orbit and choosing a new section $\tilde{\sigma}$ that picks out the individually shifted representatives. This provides an intuitive graphical interpretation of these transformations. While the standard classical, model-independent frame transformations simply shift the models on each orbit by the same constant amount, QRF transformations offer the additional flexibility to apply different translations to the models on different orbits and to choose a new section to pass through the transformed models accordingly (see Fig. 5). Note that given a general symmetry group, its gauge orbits are in general not one-dimensional, although we will still represent them as lines for simplicity. This representation is faithful only for one-dimensional groups, whose action can be depicted in a one-to-one manner along the one-dimensional orbits in our figures. In the specific case of one-dimensional groups, such as the translation group, we also have a notion of order, i.e. of acting with a larger or smaller group element. Hence for translations, but not in general, we can visualise the value of the group element as the length of $a(\varphi)$.

Let us illustrate the QRF transformation and its representation within the space of models with a concrete example. Take a massive gravitating object $M$ and place it in a spatial superposition of two locations. Add a light probe particle $P$ localised in one position, in between the locations of the massive object. This situation, which is similar to the one considered in ref. 17, is illustrated by the drawing on the far right-hand side of Fig. 6. We can now describe the scenario from the perspective of different reference frames.

In the original frame (or representational convention) $\sigma$, we take the zero-point of the coordinate system to be associated with the position of the particle $P$, as indicated by the flag in Fig. 6. Thus $\sigma$ is pictured as the horizontal dotted line. The position of the massive object is in a superposition of being on the right and on the left of the particle. For concreteness, let us assume that, in this reference frame, $M$ is at position $a$ in the former

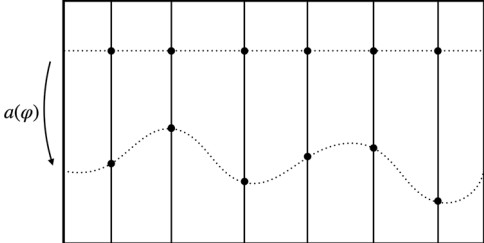

**Fig. 5 | A QRF transformation for the translation group for a superposition of several different configurations.** In general, such a transformation can apply a *different* translation $a(\varphi)$ in each branch of the superposition. In the framework of representational conventions, this is accounted for by the orbit-dependence of the group element $g_{\sigma \to \tilde{\sigma}}^{(\mathcal{O}_\varphi)}$ relating two representational conventions $\sigma$ and $\tilde{\sigma}$.

and $-a$ in the latter branch. That is, the systems are in the state

$$|\psi\rangle_{PM}^{(P)} = |0\rangle_P \otimes (\alpha|-a\rangle_M + \beta|a\rangle_M), \ \alpha, \beta \in \mathbb{C}, |\alpha|^2 + |\beta|^2 = 1, \quad (3)$$

relative to the particle. Note that the state of the particle relative to its own reference frame factorises out. This is a general feature of states given relative to QRFs: the relative state of the frame itself factorises out[8] and is therefore sometimes omitted[2]. (Note that while this is taken as a physically motivated assumption in[8], it is argued for in ref. 22 by specifying the conditions under which a frame satisfies a localisability property. Since this always holds for what we refer to as perfect reference frames (see the next section), this can be seen as additionally corroborating this assumption.) Now, if the two-particle system is described, for example, by Newtonian gravity, the overall configuration of the two systems is translation-invariant. For simplicity, we exclude the rotational symmetry of the theory and consider only the invariance under translations in one dimension. As a consequence, any two models that are related by a rigid translation live on the same orbit. This is the case for models with the same relative distance $d_{MP} = x_M - x_P$. Thus in this example, each orbit can be labelled by the relative distance between the two systems $P$ and $M$. As the (signed) relative distance between $M$ and $P$ differs across the branches in our setup, we are dealing with a superposition of two models that are physically distinguishable and therefore live on different orbits in the space of models (see Fig. 6). Note that we would consider superpositions of models living on the same orbit as "false" superpositions. In the present context, this would correspond to a superposition of two configurations with the same relative distance between $M$ and $P$ but a distinct centre of mass position.

Next, we can change to the reference frame associated with $M$, in which the zero-point of the coordinate system is associated with the position of $M$. This entails translating by different amounts $a$ and $-a$ in the different branches, respectively. The translation $a(\varphi)$ performed in the quantum frame change thus depends on the orbit $\mathcal{O}_\varphi$—it is given by the distance between $P$ and $M$ in each $\varphi$. Since rigid translations keep the relative distance between systems invariant, the probe particle is now at different distances from the zero-point in the two branches, namely at $a$ and $-a$. Finally, we need to choose a new representational convention—that is, a section $\tilde{\sigma}$ through the space of models—which includes the models in which $M$ is at the origin. Altogether, this leads to the state

$$|\psi\rangle_{MP}^{(M)} = |0\rangle_M \otimes (\alpha|a\rangle_P + \beta|-a\rangle_P), \ \alpha, \beta \in \mathbb{C}, |\alpha|^2 + |\beta|^2 = 1. \quad (4)$$

Again, note that the state of the system whose position defines the origin of the QRF (that is, the massive object $M$), factorises out. The resulting configuration is depicted on the upper right-hand side of Fig. 6; and $\tilde{\sigma}$ is the diagonal dotted line.

**Fig. 6 | The space of models for a system consisting of a massive object $M$ and a particle $P$.** This figure depicts a superposition of two distinct physical configurations—$M$ being to the left or to the right of $P$—and two different representational conventions $\sigma$ and $\tilde{\sigma}$. The straight line represents the section $\sigma$ in which the zero-point of the coordinate system is associated with the position of $P$, marked by the flag. In this reference frame, $M$ is at different positions across the branches. The tilted line represents the section $\tilde{\sigma}$ in which the zero-point is associated with the position of $M$, marked again by the flag, while $P$ is in a superposition of positions. The change from one representational convention to another is implemented by a quantum translation. The images outside the space of models illustrate the frame-dependence of the superposition.

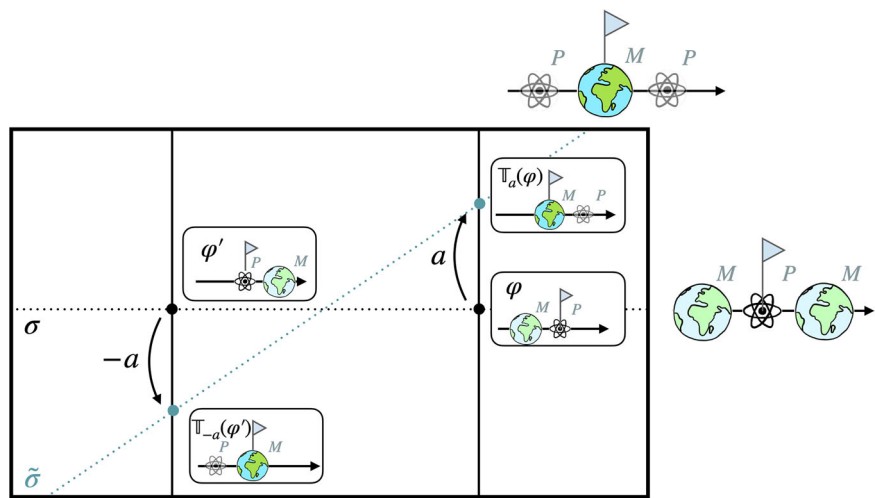

https://doi.org/10.1038/s42005-025-02084-3                                                                                    **Article**

Figure 6 also clearly illustrates one of the core features of QRF transformations: the frame-dependence of superposition[2,8]. While it is the massive object that is in a superposition of locations in the original frame of reference, this superposition is "shifted" to the particle in the new frame. Using the framework of representational conventions, we can explain this phenomenon in more detail as follows: without a representational convention, we have no way to identify locations across the different models or, in the quantum case, branches of the superposition. Whether an object is at "the same" or a "different" location depends on the frame of reference. As explained in the previous section, each representational convention defines a counterpart relation between a pair of models, which prescribes how to compare the two.

Let us consider the concrete counterpart relations between models in the different reference frames for this example. We start with the QRF of the probe particle, in which the particle is located at the origin of the coordinate system in both branches and the massive object is in a superposition of two locations, namely $a$ and $-a$. The two superposed models are illustrated in Fig. 6 as the two models lying *on* the section $\sigma$ associated with the particle. By definition, the counterpart relation of the two models lying *on* the section is the identity:

$$\mathrm{Counter}_\sigma(\varphi, \varphi') = \mathrm{Id} . \tag{5}$$

In other words, we can use the identity to compare the locations of the massive object across the two branches. Now, we change to the reference frame of $M$ in two steps. First, we apply a branch-wise translation to each of the models. This is illustrated in Fig. 6 by moving the models by $-a$ and $a$ along their orbits, respectively. Now, the counterpart relation relative to the original section $\sigma$ is no longer the identity:

$$\mathrm{Counter}_\sigma(\mathbb{T}_a\varphi, \mathbb{T}_{-a}\varphi') = -(a) + (-a) = -2a. \tag{6}$$

Thus, we need to be careful when comparing the locations of the systems across the superposition. In the second step of the QRF change, we select the new section $\tilde\sigma$, which is associated with $M$ and contains the translated models. This means that we now identify the locations of the massive object. Again, by definition, the counterpart relation of the translated models relative to the section $\tilde\sigma$ containing them is the identity element of the group,

$$\mathrm{Counter}_{\tilde\sigma}(\mathbb{T}_a\varphi, \mathbb{T}_{-a}\varphi') = \mathrm{Id}. \tag{7}$$

As a result, we can directly compare the locations of the particle across the branches in this representational convention, simply using the identity.

We make implicit use of counterpart relations whenever we speak of superposition. After all, an object is in a *superposition* of locations whenever it is in *different* locations across the different branches. Yet, when we take into account the symmetries and redundancies of a theory we need to be careful when comparing the location of objects across different branches in the superposition. Thus we have seen that whether one or the other object is in a superposition of locations can depend on the frame of reference. However, let us stress that whether quantities that are *invariant* under the action of the symmetry group are in superposition *cannot* be changed by a QRF transformation. This is because the latter only changes what a configuration is described relative to, but does not affect the physical relations between the component systems, such as relative distances, or more generally the relational properties of the component systems. In our example, the relative distance between the massive object and the probe particle in each branch is invariant under a change of QRF. As these quantities are invariant under any classical symmetry transformation, they are also not affected by the branch-dependent extension thereof[2,17].

Thus, we can maintain that while classically invariant quantities are also invariant under branch-dependent transformations, which system is in a superposed state for a general gauge-variant quantity (such as position, in our example) depends on the choice of reference frame. Similarly, whether

or not two systems are entangled can also change under QRF transformations since the latter change the factorisation of the total Hilbert space[2,8].

Quantum states relative to a chosen QRF have two defining features: (i) the reference system is not entangled with any of the other systems, and (ii) it is found in a definite state, i.e. in an eigenstate of the quantity of interest (position, in our example). It has been argued in the QRF community that a change of QRF is somehow related to a shift in the Heisenberg cut. One way to understand this is as follows: due to the fact that the reference system always factorises out in its own reference frame (cf. (ii)), its state is, in some works on QRFs, even omitted entirely from the quantum description[2]. In this sense, choosing a QRF can then be seen as shifting the reference system across the "Heisenberg cut" to the classical side of the description. However, making precise what exactly the Heisenberg cut represents in this situation, how it relates to possible measurements, and to what extent the state of the reference frame qualifies as "classical", requires an investigation that goes beyond this paper's scope.

Note that both these features, (i) and (ii), have analogues at the classical level. We already saw that the frame-dependence of superposition can be traced back to the change in how we compare different possible states of a system (or in philosophical jargon: different possible worlds)—a phenomenon that can be understood already at the classical level, although most practical implications are revealed when considering quantum superpositions. Similarly, the frame-dependence of the Hilbert space factorisation has a classical analogue in the reshuffling of the degrees of freedom through a model-dependent transformation (cf. ref. 23).

Consider the following example, which was worked out in ref. 10, of a three-particle system, whose positions are denoted by $q_1$, $q_2$, $q_3$. When applying a standard, model-independent translation, the position of each system is shifted by the same fixed amount: $q_i \rightarrow q_i + a$, $i = 1, 2, 3$. For the *model-dependent* translation, on the other hand, the amount by which the positions are shifted can depend on the model $\varphi = (q_1, q_2, q_3)$ and thus on the configuration of the other particles. This is the case, for example, when changing from the reference frame of Particle 1 into the reference frame of Particle 2, even at the classical level. In the former, what we mean by "the position of Particle 3" is "the position of Particle 3 relative to Particle 1", that is $Q_3^{(1)} = q_3 - q_1$. Upon changing into the reference frame of Particle 2, on the other hand, "the position of Particle 3" is now $Q_3^{(2)} = Q_3^{(1)} + q_1 - q_2 = q_3 - q_2$, namely "the position of Particle 3 relative to Particle 2". In this sense, this transformation "mixes" the degrees of freedom of the different subsystems and thereby changes the notion of subsystem. While this change of subsystem structure is a well-understood feature at the classical level, it develops new implications at the quantum level, where it is reflected in a refactorisation of the Hilbert space and thereby the frame-dependence of entanglement.

The frame-dependence of superposition and entanglement is not only of interest to our general understanding of these features but also has practical implications. As shown in ref. 17, within Newtonian and post-Newtonian gravity, changing to the reference frame associated with the position of the massive object allows one to predict the motion of a test particle in the presence of a gravitational source in superposition while staying agnostic about the nature of the spacetime sourced by a massive object in superposition. The idea is to assume that the equations of motion are invariant under the above-described changes of the reference frame. Given this assumption, one can simply solve the problem in the reference frame associated with the massive object itself, in which case the gravitational source is localised in a definite position and we can use standard quantum theory on a fixed curved spacetime background to determine the motion of the probe particle in superposition. This opens up a possible route to empirically test the extended symmetry principle, i.e. to check the invariance of physical laws under such changes of frame, at least in the context of ref. 17. Assuming that we had the technical abilities to place the massive object in a superposition of two different locations (see ref. 54 for a discussion of feasibility), we could let the particle fall freely in the gravitational field of the mass and compare the motion of the particle with the predictions based on quantum frame changes and the dynamics on a fixed

background. If the predictions turned out to truly describe the motion, this would give strong support to the equations of motion actually being invariant under quantum frame changes and thus to the extended symmetry principle. At the same time, this would allow one to experimentally falsify gravitational collapse models[55–57] and semi-classical gravity[58], which violate the extended symmetry principle, in this regime (see also discussion in ref. 17).

## Locally compact groups and imperfect quantum reference frames

While in the previous section we studied an example of a concrete symmetry group, namely the one-dimensional translation group acting on a two-particle system, the framework of QRFs allows for the treatment of a much larger class of systems. Let us briefly illustrate this with the help of the group-theoretic treatment of QRFs as studied in ref. 8. This general treatment is applicable to an arbitrary number $N$ of systems that all transform under a locally compact Lie group $G$, that is, they all carry some unitary representation of $G$. To set the scene, consider first a single system whose configuration space is denoted by $X$ and is typically taken to be a set with a manifold structure. (Note that we are using the terms "configuration space" and "configurations" very generally, i.e. not limited to spatial configurations.) The action of the locally compact Lie group $G$ on $X$ is taken to be transitive, that is, $\forall x, x' \in X \; \exists g \in G$ s.t. $x' = gx$. In that case, $X$ is also called a *homogeneous space*. In most cases treated in the literature (e.g. refs. 2,3,24), the group action is taken to be regular, that is, transitive *and* free. This means that $X \cong G$ as manifold and thus there exists a unique group element that relates any pair of elements of the configuration space. In other words, $G$ acts on itself via group multiplication. $X$ is then a *principal* homogeneous space for $G$. Note that the uniqueness of the group element relating any two points in $X$ will translate directly to there being a unique reference frame change between the subsystems. Now, let us take $N$ systems, each of which transforms regularly under $G$ and has configuration space $X$. Then, the *relative* configuration of all systems with respect to a chosen subsystem $i$ can be expressed as a tuple of group elements, each of which specifies the relative orientation of the system with respect to the reference system $i$. Importantly, the system $i$ is trivially related to itself with the group element $e$. Embedding the relative configurations of each of the $N$ systems in a Hilbert space $L^2(G)$, we can assign general relative states to the systems of the form $|e\rangle_i \otimes |\psi\rangle_{\bar{i}}$ where $\bar{i}$ indicates all systems except the $i$th one. Relative states of this form are also referred to as "aligned states"[12] and live in a subspace of $L^2(G)^{\otimes N}$. Note that quantum systems that carry the regular representation of $G$ and have assigned Hilbert space $L^2(G)$ are referred to as *perfect quantum reference frames*. Perfect QRFs are also referred to as "ideal" in ref. 14 and "principal" in ref. 22. Moreover, note that we take here the convention of the *left* regular representation, that is, $U(h)|g\rangle = |hg\rangle$. They are called "perfect" because the set of states $\{|g\rangle\}_{g \in G}$ are orthogonal for the state space $L^2(G)$, i.e. $\langle g|h\rangle = \delta(g^{-1}h)$, and can be perfectly distinguished. In other words, we can perfectly differentiate between two states localised in configuration space, no matter how "similar" they are.

Let us now see how this group-theoretic formulation of QRFs carries over into our framework. Given $N$ systems carrying the regular representation of $G$ as described above, each with configuration space $X$, our space $\Phi$ of (classical) models is $X^{\times N}$, where $\times$ denotes the Cartesian product. The latter can be partitioned into orbits of the group $G$ (where $G$ acts uniformly, i.e. with the same group element $g \in G$, on each of the $N$ factors in $X^{\times N}$), which can be characterised by the invariant relative configurations of the systems. Any two models that are assigned the same relative configurations are physically equivalent and thus lie in the same orbit. The choice of subsystem $i$ as a (perfect) QRF can be seen as choosing the section that, for each orbit, picks out the relative configuration $e$ for subsystem $i$. Then, a change from reference frame $i$ to reference frame $j$ is implemented by "reading off" the orbit-dependent group element that relates the new frame $j$ to the old frame $i$ and performing a quantum-controlled transformation on the state. Note that in the space of models, these group elements are exactly the $g_{\sigma_i \to \sigma_j}^{(\mathcal{O}_\varphi)}$ for each orbit $\mathcal{O}_\varphi$. The resulting model that acts as a

representative for $\mathcal{O}_\varphi$ then lies on the section that assigns the identity element $e$ to the subsystem $j$.

These perfect QRFs are associated with the rather classical idea of being able to perfectly distinguish any two states or models. In order to model reference frames more realistically, we ought to give up on this idealisation. Various different ways of modelling more realistic QRFs can be found in the literature[14]. In the following, we show how one can make sense of them within our framework. So far, mainly two options have been considered.

A first option is concerned with considering additional symmetry properties of the reference frame. As an example, let us take a clock $C$ as a reference system and a system of interest $S$. If the clock was a perfect QRF, we would assign states in $L^2(\mathbb{R}) \otimes L^2(\mathbb{R})$ to the clock and the system, that is, they would both carry the regular representation of $G = (\mathbb{R}, +)$. In the spirit of the Page–Wootters formalism[59–61], a product state of the two systems would be written as $|t\rangle_C \otimes |\psi\rangle_S$, where $|\psi\rangle_S$ is the state of the system *conditioned* on the clock showing time $t$. Now, let us imagine that we choose an imperfect clock system that carries a certain periodicity, e.g. a typical analogue clock that can only distinguish times up to 12 h, i.e. which, for any time $t$, enters at time $t + 12$ h, the very same state that it had at time $t$. Formally, this can be modelled by assigning to the clock a smaller symmetry group, in our example $H = \mathbb{Z}_{12}$. As shown in ref. 14, this means that the symmetry properties of the reference system $C$ are "shifted" onto the description of the system $S$. Within our framework, any two states $|t\rangle \otimes |\psi\rangle_S$ and $|t + 12m\rangle \otimes |\psi\rangle_S$, $m \in \mathbb{Z}$ would be identified with one another. In more technical terms, this additional invariance is captured by the existence of a non-trivial stabiliser group $H_x = \{h \in G | h \cdot x = x\}$ for $x \in X$, i.e. a subgroup that leaves some configuration $x$ of the reference system unchanged under the group action. A very simple example is the group of rotations about the origin in $\mathbb{R}^2$, i.e. the group $S^1$. The origin is fixed by every rotation, so its stabiliser group is all of $S^1$. The action of $G$ on the reference frame is thus only transitive but not free. A QRF that has a non-trivial stabiliser group is one instance of an imperfect QRF. Such an imperfect QRF is referred to as an "incomplete" frame in refs. 14,22.

The existence of stabilisers is also discussed in ref. 25 in the context of the space $\Phi$ of models of general relativity. Note that, there, only a subset of the orbits carry the additional symmetry of the stabiliser group, leading to orbits of various "sizes". In fact, this means that the space of models can be described as an orbifold (cf. ref. 50). As a consequence, the counterpart relation becomes ambiguous. We will discuss the implications of stabilisers in more detail in the context of spacetimes in superposition below.

A second option to extend beyond perfect QRFs is concerned with the implicit assumption that we can distinguish arbitrarily well between two models, no matter how similar their configurations are. In fact, for perfect frames, any two states of different models will be orthogonal. Dropping this assumption, we can instead choose a *coherent state system* $\{|\phi(g)\rangle = U(g)|\phi(e)\rangle_{g \in G}\}$, with $U(g)$ a unitary representation, as a basis for the Hilbert space of the reference frame rather than the orthonormal basis $\{|g\rangle \in L^2(G)\}_{g \in G}$[14]. These coherent states have to satisfy the resolution of the identity $\int dg |\phi(g)\rangle\langle\phi(g)| = n \cdot \mathbb{1}$, where $n > 0$ is a normalisation factor that is commonly set to one; but they do not necessarily satisfy $\langle\phi(g)|\phi(h)\rangle = \delta(g^{-1}h)$. Since two different coherent states can have non-zero overlap, the same holds for the states of two different models. This encodes the idea that the states assigned to models can be "blurred out" and can thus not be resolved beyond some fundamental uncertainty. In[7,14,62,63], such imperfect QRF, commonly referred to as "non-ideal", are treated. But so far, the QRF approach of refs. 2,5,8,11,17,18 has not been extended to include coherent states. So in the following, we will thus focus on perfect quantum reference frames, that is, those that carry the regular representation of the symmetry group. In particular, this implies that the action of the group is taken to be transitive and free. We leave to future work the development of the present framework to incorporate non-ideal QRFs.

Before closing this section, let us return to the case of perfect frames and the imperfect ones discussed in the first option above. In this case, a choice of reference frame can be seen as a choice of the section, in which the state of

the reference system is set to a fixed configuration, in the above case the trivial state $|e\rangle$, and thereby factorises out. Viewing choices of QRF in this more abstract way (see also discussion in ref. 18) allows us to embed a large number of earlier works on QRFs in our framework: the translation group which has been our case study presented above[2], the spin rotation group[5,11], the Lorentz group[20], the conformal symmetries[18], and the (not locally compact) quantum diffeomorphisms[19], which will be the subject of the next subsection.

### Quantum coordinate fields and quantum reference frame changes

Finally, let us turn to the example of general relativity with the symmetry group taken to be the diffeomorphism group. This case is of particular interest because of its potential implications for a theory of quantum gravity. Clarifying the role of coordinates, symmetries, and reference frames in classical general relativity was central in its development and is still a subject of active debate in the philosophical literature about various incarnations of the *hole argument* (see refs. 30,64 for recent reviews). This clarity is crucial when trying to generalise these ideas to the quantum level, where many of the conceptual difficulties are exacerbated. In this and the following section, we analyse *quantum diffeomorphisms* (variants of which have been proposed in refs. 19,39,65) and their conceptual implications by embedding them in the space of models. Moreover, we propose a possible strategy for defining a counterpart relation, i.e. a prescription of what counts as "identity" for points in different spacetimes in superposition, from the fields living thereon. The more philosophically oriented reader may want to note that whenever we use "identity" or "identify" in the following, we do not mean it in the strict sense of items that are utterly identical but rather in the weaker sense of Lewis' counterpart relation[27–29] or the "threading" of Gomes and Butterfield[25]. Based on this, we will show how to construct a change of QRF for the diffeomorphism group, in the sense of a change of representational convention. This provides the foundation for a new perspective on what one may call a *quantum hole argument* (cf. ref. 66)—that is, an extension of the classical hole argument to spacetimes in superposition—which will be explored in the next section.

In vacuum general relativity, a model is defined by a configuration $(\mathcal{M}, g_{ab})$ of a manifold $\mathcal{M}$ and a Lorentzian metric $g_{ab}$. While we could, in full general relativity, consider any type of matter fields in addition to the gravitational degrees of freedom, we will first focus our attention on sets of four smooth scalar fields $\{\chi_{(A)}\}_{A=0,1,2,3}$. These scalar fields will later play an important role in the construction of what we will call the *comparison map* and, in particular, a choice of a set of four such scalar fields will act as a reference frame. Let us therefore briefly comment on their interpretation. We have essentially three options[67].

First, we can see them as idealised or coordinated fields, whose dynamics and backreaction on the curved spacetime can be neglected for all practical purposes. That is, in this approximation, the fields do not feel the influence of the metric, and do not influence the spacetime itself. This choice has the advantage that for any physical problem, we are free to identify a suitable coordinate system, since the reference fields are not tied to a physically instantiated system. For instance, we could postulate an atlas without further elaboration on the dynamics of the coordinate fields or, alternatively, we could have coordinate fields that obey dynamical laws governed by something other than the spacetime metric, e.g. $\Box_\eta \chi_{(A)} = 0$, for $\eta$ an auxiliary metric.

A second option is to model the scalar fields as dynamical fields that "feel" the influence of the metric but do not backreact on the latter. As an example, we could take the four scalar fields $\{\chi_{(A)}\}_{A=0,1,2,3}$ to be solutions to the Klein–Gordon equation in curved spacetime. This approach is advantageous as it represents the fields in a more realistic manner, lending more physical significance to relational quantities expressed relative to these fields.

As a third option, we can incorporate the backreaction of the reference fields. Although this aligns with the most realistic way of modelling the

scalar fields, it requires solving the non-linear Einstein equations in their entirety. While this is possible for simple models, it restricts the freedom in the choice of reference fields drastically.

The arguments that follow in this section do not rely on choosing any particular one of these three options. Returning to the formal setup, let us now define a model as the tuple $(\mathcal{M}, g_{ab}, \chi_{(A)}, \tilde{\chi}_{(A)})$, consisting of a manifold, a metric, and two sets of reference fields, whose respective perspectives we are going to adopt below. The set of all these kinematically possible models is the space of models $\Phi$. The diffeomorphism group, $G = \text{Diff}(\mathcal{M})$, partitions it into orbits of physically equivalent models, which can be related to one another by a diffeomorphism. As before, we take the conceptual leap to the quantum realm by considering superpositions of different configurations. This corresponds to a situation in which the metric and the scalar fields have quantum properties in the sense that they can be in different configurations $g_{ab}^{(i)}$, $\chi_{(A)}^{(i)}$, and $\tilde{\chi}_{(A)}^{(i)}$ in each branch $i$ (see Fig. 7). For clarity, we also denote the manifold in each branch by $\mathcal{M}_i$, even though all $\mathcal{M}_i$ as differentiable manifolds are assumed to be diffeomorphic, and are, strictly speaking, part of the theory and not of each individual model. In the following, we will focus on the case $i = 1, 2$; in general, however, we can consider superpositions of an arbitrary number of models.

Regarding the Hilbert space structure of the gravitational field, we want to keep our assumptions minimal. Let us emphasise that we do not claim to provide a full construction of the Hilbert space for the metric. In particular, we do not resolve the mathematical challenges surrounding the existence of a well-defined measure over the set of all spacetime metrics. While we recognise this as a significant open problem, our focus here lies on the conceptual insights that can be gained in the meantime. What we need for the present framework to apply is that the structure for the quantum description of the gravitational field reflects the symmetry properties of the theory, i.e. can be partitioned into orbits of the diffeomorphism group, and allows for some type of semi-classical states, each peaked around a classical configuration $g_{ab}^{(i)}$ of the metric (see e.g. ref. 68) and entangled with the corresponding matter field configurations $\chi_{(A)}^{(i)}$ and $\tilde{\chi}_{(A)}^{(i)}$. We would expect such states to arise in the transition from a full theory of quantum gravity to general relativity in a regime where the quantum fluctuations of the gravitational field can be neglected while the coherence of the superposition is still maintained. An example would be the scenario described above in the context of QRFs for the translation group, in which a massive object in superposition acts as the source for the gravitational field. If the linearity of quantum theory still applies in this context, the spacetime sourced by such a massive object should be in a superposition of macroscopically distinct gravitational fields: one field with the source further to the left, and the other with the source further to the right. From a mathematical point of view, all this implies is that we can consider linear combinations of the aforementioned semi-classical states in the abovementioned Hilbert space for the spacetime metric. In

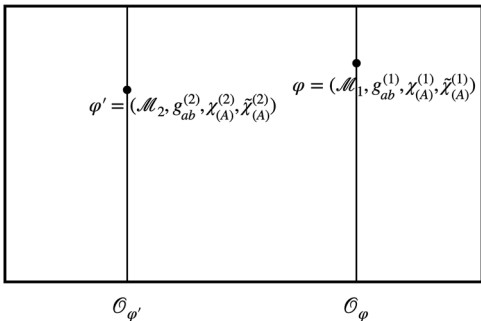

**Fig. 7 | The space of models $\Phi$ for the diffeomorphism group $G = \text{Diff}(\mathcal{M})$.** A model in this context is given by a spacetime manifold $\mathcal{M}$ with a metric $g_{ab}$ and two types of reference fields $\chi$ and $\tilde{\chi}$. The models on a single orbit are related by diffeomorphisms and are taken to represent the same physical situation. Two diffeomorphically inequivalent models, such as $\varphi$ and $\varphi'$, live on different orbits and are physically distinct.

the following, we will further limit our discussion to superpositions of physically *in*equivalent configurations $(\mathcal{M}, g_{ab}^{(i)}, \chi_{(A)}^{(i)}, \tilde{\chi}_{(A)}^{(i)})$, that is, we will not consider superpositions of models living on the same orbit.

Let us now inspect what this setup looks like as regards different choices of reference frames. We will begin by presenting the reference frame given by the $\chi$-fields and then perform the QRF change to the $\tilde{\chi}$-fields.

**Reference frame of $\chi$.** In analogy to the earlier example of a massive object in superposition, which we considered from the perspectives of the Earth and the probe particle respectively, we now want to describe a superposition of spacetimes with respect to the $\chi$- or $\tilde{\chi}$-fields respectively. But what does it mean to go into the reference frame associated with, say, the $\chi$-fields precisely? Intuitively, this means using the values of the four scalar fields $\chi_{(A)}$ to label the points and, in particular, "identify" them across the different branches in superposition. Just as we used the Earth or the probe particle to define the zero-point of our one-dimensional system across the different branches, we now use the values of the four scalar fields $\chi_{(A)}$ to label the points of the four-dimensional spacetime manifold. And just as associating the zero-point to a particular particle allowed us to sew together the different branches of the superposition, thus defining a counterpart relation that tells us whether two locations are the same or different across the branches, we can use the values of the $\chi$-fields to identify points across the spacetimes in superposition and thereby represent different field configurations as "inhabiting" the same spacetime points.

To be more precise, we can use the $\chi$-fields to (i) fix a representational convention and (ii) define the associated counterpart relation (see Fig. 8). Firstly, in order to fix a representational convention, we need to pick out one unique model for each orbit. If the four scalar fields $\{\chi_{(A)}\}_{A=0,1,2,3}$ define a bijective map from $\mathcal{M}$ to $\mathbb{R}^4$, fixing the values of the fields removes the redundancy induced by the diffeomorphism invariance. Note that fixing the position of a single particle, as suggested in refs. [33,34], will not suffice to single out a unique representative for each orbit but rather restricts one to a —still infinite-dimensional—set of possible representatives compatible with the chosen position for the particle. Agreed, given a specific set of scalar fields, especially if we choose to model them more realistically, the $\chi$-fields might repeat their values, so that there is no bijection. In this case, we can still work with an open subregion of $\mathcal{M}$ on which the fields define bijective maps, as is commonly done when defining coordinate fields. Nevertheless, to keep our notation simple, we will continue referring to all of $\mathcal{M}$ in the following. Note further that there are special cases, such as when the $\chi$-fields take on periodic values over time or are homogeneous over a region of space (cf. ref. [69]), for which the non-uniqueness arises from the existence of stabilisers, and thus imperfect QRFs, as discussed at the end of the "Locally compact groups and imperfect quantum reference frames" subsection.

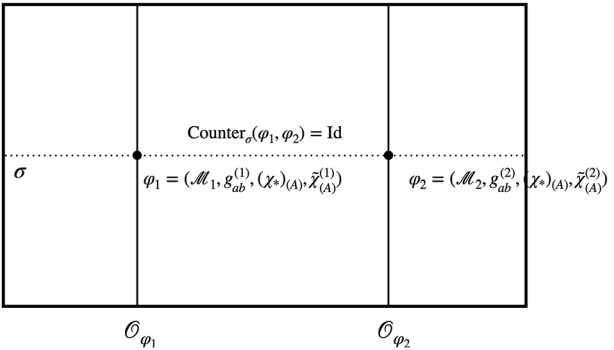

**Fig. 8 | Reference frame of $\chi$: superposition of two models with the same configuration $\chi_*$ for the reference fields.** In this reference frame, the comparison map $C_\chi$ is defined through $\chi$—that is, we identify points in $\mathcal{M}_1$ and $\mathcal{M}_2$ if and only if the values of the $\chi$-fields are the same at those points. The comparison map corresponds to the counterpart relation $\mathrm{Counter}_\sigma(\varphi_1, \varphi_2)$ for the section $\sigma$. Because the reference fields are in the same configuration $\chi_*$ across the different branches, it is equal to the identity.

More precisely, in that case, the redundancy is not lifted entirely and it is not possible to pick a single representative of the orbit. As a consequence, the characterisation and identification of points across a superposition of spacetimes via coincidences of field values become non-unique.

Putting aside these qualifications, let us assume for the remainder of this section that the fields are inhomogeneous enough to define bijective maps from $\mathcal{M}$ to $\mathbb{R}^4$. In this case, fixing the configuration of the $\chi$-fields in each model defines a section through the space of models. Let us denote by $\chi_*$ one particular such field configuration. While the particular choice of $\chi_*$ is a mere matter of convention, it is important that, in the reference frame associated with the $\chi$-fields, they take the same value across all models, i.e. $\chi^{(1)} = \chi^{(2)} = \chi_*$ in the case of a superposition of two branches. This allows us to describe the quantum state associated with a superposition of such models in the form of a tensor product between the state of the $\chi$-fields, which are in the definite configuration $\chi_*$, and the state describing all other fields, such as the metric. Formally, we can write this as

$$|\psi\rangle^{(\chi)} = |\chi_*\rangle_\chi \otimes \left( \alpha |g^{(1)}\rangle_g |\tilde{\chi}^{(1)}\rangle_{\tilde{\chi}} + \beta |g^{(2)}\rangle_g |\tilde{\chi}^{(2)}\rangle_{\tilde{\chi}} \right),$$
$$\alpha, \beta \in \mathbb{C}, \ |\alpha|^2 + |\beta|^2 = 1. \tag{8}$$

Here, a state $|g\rangle$ assigned to the metric should be understood *formally* as a semi-classical state peaked around a particular configuration of the metric. In the present work, we are interested in the conceptual implications that should hold in any theory of quantum gravity that satisfies our assumptions, that is, one that allows for superpositions of such states and for an action of the diffeomorphism group on a configuration $|g\rangle$ such that $U_d|g\rangle = |d_*g\rangle$. As before, expressing the state in the form of Eq. (8) could be seen as moving the $\chi$-fields across the "Heisenberg cut", which allows us to treat them classically within this specific choice of description. This is what it means to be in the reference frame of $\chi$.

Moreover, we can use the $\chi$-fields to define a natural counterpart relation, which allows us to compare points across different spacetimes in superposition. To see how, let us take a step back and consider two different configurations $(\mathcal{M}_1, g_{ab}^{(1)}, \chi_{(A)}^{(1)}, \tilde{\chi}_{(A)}^{(1)})$ and $(\mathcal{M}_2, g_{ab}^{(2)}, \chi_{(A)}^{(2)}, \tilde{\chi}_{(A)}^{(2)})$ as depicted in Fig. 9, not yet fixing the field configurations $\chi^{(1)}$ and $\chi^{(2)}$ to be the same. Now, each point $p \in \mathcal{M}_1$ is associated to a set of four values in $\mathbb{R}^4$ through $\chi^{(1)}$. Similarly, $\chi^{(2)}$ assigns a unique set of four values to each point $q \in \mathcal{M}_2$. If we take $\chi^{(1)}$ and $\chi^{(2)}$ to be merely different configurations of the same physical field, this suggests a natural strategy for identifying points across the manifolds $\mathcal{M}_1$ and $\mathcal{M}_2$: we call the points $p \in \mathcal{M}_1$ and $q \in \mathcal{M}_2$ identical— that is: we identify them, or say that they are counterparts—if and only if the field $\chi$ takes the same value at these points.

More precisely, we identify $p$ and $q$ if and only if $\chi^{(1)}(p) = \chi^{(2)}(q)$. This way of identifying points across different models is inspired, in part, by the work of Westman and Sonego[70], who construct a *space of point-coincidences*, that is the collection of joint readings $(\Phi_1, \ldots, \Phi_N)$ of a sufficient number of fields $\Phi_N, N \in \mathbb{N}$ that generically (excluding too homogeneous configurations) reproduces the spacetime manifold. We can formalise this by composing the maps into an identification or *comparison map*

$$C_\chi \equiv (\chi^{(2)})^{-1} \circ \chi^{(1)} : \mathcal{M}_1 \to \mathcal{M}_2. \tag{9}$$

This map associates to each point $p \in \mathcal{M}_1$ a unique counterpart $q = C_\chi(p) \in \mathcal{M}_2$ and thus provides us with one way of specifying what we mean by "the same" or "different" points across the different spacetimes (see Fig. 9). It thus furnishes a concrete, physically motivated realisation of the counterpart relation of Gomes and Butterfield[25]; cf. the discussion of Eq. (2) above. Alternative realisations include the Kretschmann–Komar coordinates[71,72], which are based on gravitational degrees of freedom, Geroch's use of tetrad fields to define limits of spacetimes[40], and Barbour's treatment of best-matching[73]. In addition, the identification of spacetime points across different spacetimes in superposition has been extensively discussed by Hardy[39] as well as Jia[41].

**Fig. 9 | The comparison map $C_\chi$.** Consider two configurations $(\mathcal{M}_1, g_{ab}^{(1)}, \chi_{(A)}^{(1)})$ and $(\mathcal{M}_2, g_{ab}^{(2)}, \chi_{(A)}^{(2)})$. Each point $p \in \mathcal{M}_1$ is associated a coordinate value through the field $\chi^{(1)}$. Similarly, $\chi^{(2)}$ assigns a coordinate value to each point $q \in \mathcal{M}_2$. The comparison map $C_\chi = (\chi^{(2)})^{-1} \circ \chi^{(1)}$ maps the point $p \in \mathcal{M}_1$ to $q \in \mathcal{M}_2$ if and only if $\chi^{(1)}(p) = \chi^{(2)}(q)$.

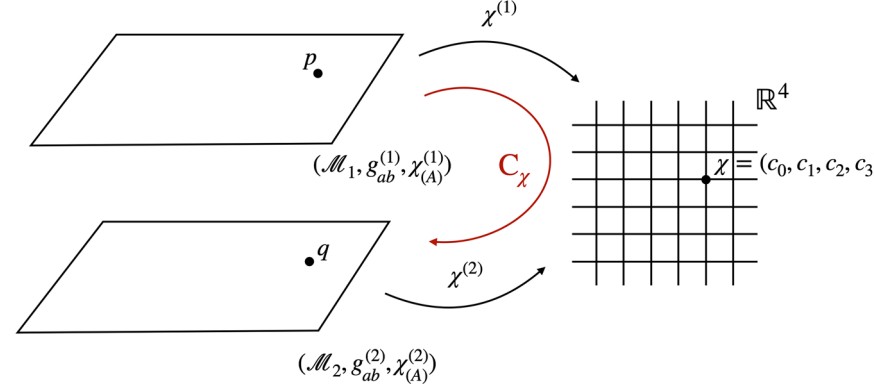

**Fig. 10 | A QRF transformation for the diffeomorphism group.** It consists of two steps: **a** the application of a quantum diffeomorphism and **b** a change of representational convention to align with the new models. While the former transforms the models, leaving the section $\sigma$ the same, the latter changes the section from $\sigma$ to $\tilde{\sigma}$ and thereby changes the counterpart relation between any two given models.

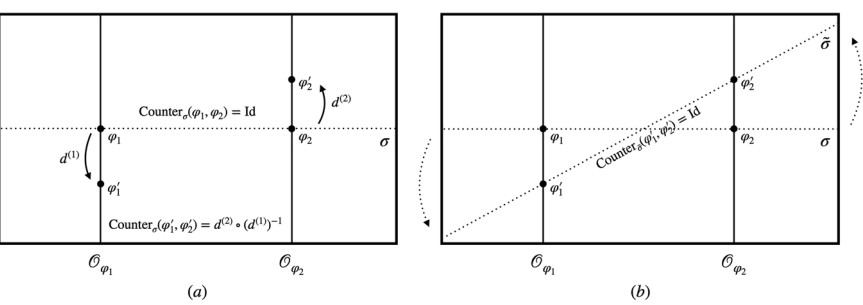

Returning to the comparison map, given a superposition of two models $\varphi_1 = (\mathcal{M}_1, g_{ab}^{(1)}, \chi_{(A)}^{(1)}, \tilde{\chi}_{(A)}^{(1)})$ and $\varphi_2 = (\mathcal{M}_2, g_{ab}^{(2)}, \chi_{(A)}^{(2)}, \tilde{\chi}_{(A)}^{(2)})$, $C_\chi$ can be shown to be equivalent to the counterpart relation for the section $\sigma$ on which the $\chi$-fields take the same value $\chi_*$. Denote by $d^{(1)}$ and $d^{(2)}$ the diffeomorphisms such that $\chi_* = \chi^{(1)} \circ (d^{(1)})^{-1} = \chi^{(2)} \circ (d^{(2)})^{-1}$. (Note that the action of the diffeomorphism on the fields is the inverse of its action on manifold points—we can see the action of a diffeomorphism $d$ either as moving the points as $p \to d(p)$ or as shifting the fields inversely as $\chi \to d_*^{-1}(\chi)$.) Then, the counterpart relation is given by

$$\text{Counter}_\sigma(\varphi_1, \varphi_2) = (d^{(2)})^{-1} \circ d^{(1)} = (\chi^{(2)})^{-1} \circ \chi_* \circ (\chi_*)^{-1} \circ \chi^{(1)} = C_\chi. \tag{10}$$

where the first equation follows from the definition of "Counter" in Eq. (2). Moreover, we can now see in what sense this defines a natural way of comparing points across different spacetimes in superposition. In the reference frame of $\chi$, the $\chi$-fields take the same value $\chi_*$ across all the different branches. Thus, the comparison map becomes

$$C_\chi = \chi_*^{-1} \circ \chi_* = \text{Id} \tag{11}$$

and in this way, we can simply use the identity to compare points across the different spacetimes in superposition.

While it may seem unnecessary to go through the trouble of explicitly constructing the comparison map if it ends up being the identity after all, we will see the benefit of these definitions in the "Conceptual implications of QRF transformations" section. For it is precisely the confusion around how to compare spacetime points across different possible worlds that lies at the heart of the quantum hole argument. Keeping track of how the comparison map changes under a QRF transformation will thus be crucial in the next section.

**Changing to the reference frame of $\tilde{\chi}$.** Next, let us go through the steps required to change into a different reference frame —the reference frame

associated with the $\tilde{\chi}$-fields. As we established above, a QRF transformation consists of two steps: (i) apply a quantum-controlled symmetry transformation to "re-align" the reference frame across the branches and (ii) change the representational convention such that it passes through the new models. This is depicted in Fig. 10. In the context of a diffeomorphism-invariant theory, (i) is implemented through a *quantum diffeomorphism*. Variants of these transformations have been considered before by Anandan[65], Hardy[39], and some of us[19]. They consist of applying a different diffeomorphism

$$\begin{aligned} d^{(1)} : \mathcal{M}_1 &\to \mathcal{M}_1, \\ d^{(2)} : \mathcal{M}_2 &\to \mathcal{M}_2 \end{aligned} \tag{12}$$

in each branch of the superposition respectively (see Fig. 10a).

Now, the key idea is that in order to change into the reference frame associated with the $\tilde{\chi}$-fields, we want to choose the diffeomorphisms such that $(d^{(1)})_*^{-1}(\tilde{\chi}^{(1)}) = (d^{(2)})_*^{-1}(\tilde{\chi}^{(2)}) = \tilde{\chi}_*$, fixing the fields to be in one particular configuration $\tilde{\chi}_*$ across all branches of the superposition. Note that the quantum diffeomorphism acts on all fields, including the metric and the original reference fields. Thus, the comparison map $C_\chi = \text{Counter}_\sigma(\varphi_1, \varphi_2)$ is also affected by the transformation. To see how, note that the action of the diffeomorphism on the scalar fields is given by the pullback, which acts as $\chi^{(i)} \to \chi'^{(i)}$ with $\chi^{(i)} = d_*^{(i)}(\chi'^{(i)}) = \chi'^{(i)} \circ d^{(i)}$. Inverting this relation, we have $\chi'^{(i)} = \chi^{(i)} \circ (d^{(i)})^{-1}$. Thus, the quantum diffeomorphism changes the comparison map to

$$\begin{aligned} C_\chi' &= \left(\chi^{(2)} \circ (d^{(2)})^{-1}\right)^{-1} \circ \left(\chi^{(1)} \circ (d^{(1)})^{-1}\right) = d^{(2)} \circ (\chi^{(2)})^{-1} \circ \chi^{(1)} \circ (d^{(1)})^{-1} \\ &= d^{(2)} \circ C_\chi \circ (d^{(1)})^{-1}, \end{aligned} \tag{13}$$

which is equivalent to the counterpart relation evaluated for the new models $\varphi_i' = \varphi_i^{d^{(i)}}$, that is $C_\chi' = \text{Counter}_\sigma(\varphi_1', \varphi_2')$. Eq. (13) generalises Eq. (11) of

Section 3.4 in ref. 25. In particular, it expresses the idea that the counterpart relation transforms by conjugation when applying the same diffeomorphism $d^{(1)} = d^{(2)}$ in all branches. This transformation was also noted by Hardy (see Eq. (50) in ref. 39), who considers a general identification map—not necessarily related to any physical fields or properties—as a central part of a quantum coordinate system. For completeness, let us also state the explicit form of the transformed model

$$\varphi'_i = \varphi_i^{d^{(i)}} = (\mathcal{M}_i, (d^{(i)})^{-1}_* g_{ab}^{(i)}, (d^{(i)})^{-1}_* \chi_{(A)}^{(i)}, (d^{(i)})^{-1}_* \tilde{\chi}_{(A)}^{(i)}). \quad (14)$$

It is worth noting that, following the quantum diffeomorphism, $\text{Counter}_\sigma(\varphi'_1, \varphi'_2)$ no longer coincides with the identity map, as the models are now aligned with respect to the $\tilde{\chi}$-fields rather than the $\chi$-fields.

The second step of the QRF transformation (ii) consists of choosing a new representational convention, which passes through the transformed models, and thereby picking out a new counterpart relation to compare objects across the superposition. This is illustrated in Fig. 10b. Changing the representational convention does not affect the configuration of the metric and matter fields but it changes the way we compare the spacetimes in superposition. Physically, it means that we now use the $\tilde{\chi}$-fields instead of the $\chi$-fields to label and identify spacetime points across the branches of the superposition. That is, we call two points identical—i.e. we identify them or say that they are counterparts—if and only if the $\tilde{\chi}$-fields take the same value at these points. This is implemented through the new comparison map

$$C_{\tilde{\chi}} \equiv (\tilde{\chi}^{(2)})^{-1} \circ \tilde{\chi}^{(1)} : \mathcal{M}_1 \to \mathcal{M}_2, \quad (15)$$

which, for a given superposition of two models $\varphi_1 = (\mathcal{M}_1, g_{ab}^{(1)}, \chi_{(A)}^{(1)}, \tilde{\chi}_{(A)}^{(1)})$ and $\varphi_2 = (\mathcal{M}_2, g_{ab}^{(2)}, \chi_{(A)}^{(2)}, \tilde{\chi}_{(A)}^{(2)})$, is equivalent to the counterpart relation $\text{Counter}_{\tilde{\sigma}}(\varphi_1, \varphi_2)$ with respect to the section $\tilde{\sigma}$ associated to $\tilde{\chi}$. If we evaluate this comparison map for the configurations $\varphi'_i = \varphi_i^{d^{(i)}}$ obtained through the quantum diffeomorphism, we find that the comparison map simplifies significantly. Because we have chosen the diffeomorphisms $d^{(i)}$ such that the new reference field takes the same value across all branches, the comparison map simply becomes

$$C_{\tilde{\chi}} = \tilde{\chi}_*^{-1} \circ \tilde{\chi}_* = \text{Id}. \quad (16)$$

We are thus justified to use the identity to compare points across the transformed spacetimes in superposition.

It is important to stress here the distinction between the application of a quantum diffeomorphism and a full QRF transformation. In particular, the direct comparison of spacetime points using the identity would be inappropriate, if we had just applied the quantum diffeomorphism without additionally changing the section. Because the reference fields, which are used to identify points across the spacetimes in superposition, *change* under a quantum diffeomorphism, the comparison map changes, too. This is what we observed in Eq. (13). It is only through changing the physical fields with which we label and identify the points from $\chi$ to $\tilde{\chi}$ that we regain the ability to directly compare the spacetime points across the transformed configurations. More generally, a quantum diffeomorphism can be seen as a transformation that moves the metric and any additional fields on the spacetime manifold with respect to an *external* reference frame. A full QRF transformation, on the other hand, changes between descriptions with respect to different *internal* reference frames (referred to as *dynamical* in e.g. ref. 74), that is, reference frames that are associated with a subsystem and are thus part of the model under consideration[23]. It thus also resonates more closely with the way reference frames are employed e.g. in Loop Quantum Gravity or Group Field Theory. There, in the absence of any external reference structures, one has to turn to material reference frames such as scalar fields or dust fields so as to describe the cosmological evolution (see e.g. refs. 75–77).

**Reference frame of $\tilde{\chi}$.** To conclude this section, let us summarise the description in the new reference frame $\tilde{\chi}$ (Fig. 11). We now have a

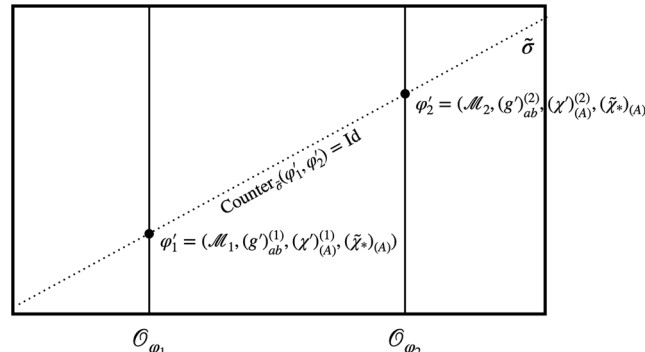

**Fig. 11 | Reference frame of $\chi$ with the same configuration $\chi_*$ for the reference fields.** The original reference fields $\chi$ may now be in a superposition of different configurations $(\chi')^{(1)}$ and $(\chi')^{(2)}$. In this reference frame, the comparison map $C_{\tilde{\chi}}$ defined through $\tilde{\chi}$, which equals the counterpart relation $\text{Counter}_{\tilde{\sigma}}(\varphi'_1, \varphi'_2)$ for the section $\tilde{\sigma}$, is the identity.

superposition of configurations $\varphi'_1 = (\mathcal{M}_1, (g'_{ab})^{(1)}, (\chi'_{(A)})^{(1)}, (\tilde{\chi}_*)_{(A)})$ and $\varphi'_2 = (\mathcal{M}_2, (g'_{ab})^{(2)}, (\chi'_{(A)})^{(2)}, (\tilde{\chi}_*)_{(A)})$. The $\tilde{\chi}$-fields now take the same value across all branches while the original reference fields $\chi$ are in superposition. The quantum state thus takes the form of a tensor product between the state of the $\tilde{\chi}$-fields in configuration $\tilde{\chi}_*$, and an, in general entangled, state describing the metric and the old reference fields:

$$|\psi\rangle^{(\tilde{\chi})} = |\tilde{\chi}_*\rangle_{\tilde{\chi}} \otimes \left( \alpha |(g')^{(1)}\rangle_g |(\chi')^{(1)}\rangle_\chi + \beta |(g')^{(2)}\rangle_g |(\chi')^{(2)}\rangle_\chi \right), \alpha, \beta \in \mathbb{C}. \quad (17)$$

Moreover, the new reference fields $\tilde{\chi}$ define how to compare points across the spacetimes in superposition through the comparison map $C_{\tilde{\chi}} = (\tilde{\chi}^{(2)})^{-1} \circ \tilde{\chi}^{(1)}$. However, since they take the same value $\tilde{\chi}_*$ in all branches of the superposition, this is simply the identity. We have thus changed to a description, which naturally implements the idea of labelling and identifying points through the $\tilde{\chi}$-fields—the reference frame of $\tilde{\chi}$.

## Conceptual implications of QRF transformations

Understanding clearly how to compare points across different spacetimes in superposition enables us now to comment on several topics, in the following subsections: first, what has been called the *quantum hole argument* (cf. ref. 66), the localisation of events, in particular in the context of interference experiments, such as the Bose et al.–Marletto–Vedral (BMV) proposal[78,79], the transformation of relational observables under QRF changes, and finally, the relation to indefinite causal order[42,43].

**The quantum hole argument.** Classically, one can cast the modern version of the hole argument, as formulated by Stachel, Earman, and Norton, as follows (cf. refs. 64,30 for reviews). Take two solutions $(\mathcal{M}, g_{ab}, \Psi_{\text{matter}})$ and $(\mathcal{M}', g'_{ab}, \Psi'_{\text{matter}})$ of general relativity, which are related by a diffeomorphism and differ only in a bounded region of spacetime (the "hole"). Because they differ only inside the hole, both are restricted to give the same configurations outside the hole. If one takes them to represent distinct physical possibilities, one has indeterminism—given a full set of initial conditions outside the hole, the spacetime and its material content can evolve into at least two distinct possibilities. A common (albeit, of course, not the only) response to this dilemma is to deny the assumption that the two solutions represent distinct physical possibilities. That is, one instead treats them as representations of the same physical situation. As a result, what apparently distinguishes the two situations—the spacetime points inside the hole, which are reshuffled by the diffeomorphism relating the two solutions—cannot have any independent physical meaning. Thus, one ends up with an argument against spacetime substantivalism, that is, against viewing spacetime as "a

substance: a thing that exists independently of the processes occurring within it"[64] (cf. also ref. 30).

This is the classical story. We now want to investigate what happens when we replace the classical diffeomorphisms with QRF transformations for the diffeomorphism group. This is interesting because, while researchers in general relativity generally endorse the idea that diffeomorphism-related configurations represent the same physical scenario, this is not uncontroversially accepted when considering scenarios involving superpositions. Specifically, there are differing intuitions regarding whether a localised test particle in the presence of a massive object in superposition represents the same physical situation as a particle in a superposition on a definite spacetime background (e.g. refs. 55–57,80)—even though these two configurations are connected by a QRF transformation (cf. Fig. 6).

In the following, let us assume that, whatever a future quantum theory of gravity may look like, it allows for a superposition of semi-classical spacetimes as discussed in the previous section; and that the laws of physics are invariant not only under classical diffeomorphisms but also under QRF transformations, i.e. changes of representational convention. As we have learned from the classical hole argument, the diffeomorphism invariance of general relativity calls into question the physical meaning of spacetime points. We can fix the descriptive redundancy originating from the diffeomorphism invariance through a choice of QRF, leading to a diffeomorphism-invariant description relative to the chosen reference system. Additionally, this choice fixes a preferred comparison map (cf. Eq. (9)) which tells us how to compare points across the different semi-classical spacetimes in superposition. If the models containing the metric and matter fields are related by a diffeomorphism, the comparison map is independent of the choice of QRF—it is, as we have seen above, always given by the diffeomorphism relating the two. Note that this is what some of us refer to as the "grain of truth in the drag-along response" to the hole argument[30].

However, when comparing models that lie on different orbits, i.e. that are not mapped into one another by any diffeomorphism, there is an additional ambiguity that comes from the choice of QRF. In particular, this is true even when using coincidences of physical field values to characterise the spacetime points, in the spirit of Einstein's "point-coincidence" argument[70,81]. For even when using such a gauge-invariant characterisation of spacetime points, there is still the question of *which* fields to use in order to define the coincidences. This corresponds to the freedom of choosing either the $\chi-$ or the $\tilde{\chi}-$ fields as QRFs in the "Quantum coordinate fields and quantum reference frame changes" subsection above.

This has important implications for the localisation of points in a superposition of diffeomorphically inequivalent spacetimes. Let us define that a pair $(p, q)$ of points $p \in \mathcal{M}_1$ and $q \in \mathcal{M}_2$ is *localised* with respect to a given comparison map $C_\chi$ if and only if $q = C_\chi(p)$. Let us now see what happens to such a localised pair of points under a QRF transformation. In the first step (i), we apply a quantum diffeomorphism, mapping $(p, q) \rightarrow (d^{(1)}(p), d^{(2)}(q))$. Note that this does not yet change the localisation since the comparison map, too, changes under a quantum diffeomorphism. In particular, Eq. (13) implies that $d^{(2)}(q) = C'_\chi(d^{(1)}(p))$ (see Fig. 12).

Now, in the second step (ii) of the QRF transformation, we change the comparison map from $C_\chi \rightarrow C_{\tilde{\chi}}$. Crucially, this transformation can take a localised pair of points into a delocalised one, in the sense that $d^{(2)}(q) \neq C_{\tilde{\chi}}(d^{(1)}(p))$. This follows directly from the fact that, in general, $C_{\tilde{\chi}} \neq C'_\chi$.

To illustrate this observation, let us consider a concrete example. The idea here is to take four scalars $R_{(A)}$, $A = 1, \ldots, 4$, formed by certain real scalar functions of the Riemann tensor $R^a{}_{bcd}$ (cf. refs. 71,72 and ref. 32,Sec. 5.1.b). Komar[72] finds these real scalars through an eigenvalue problem:

$$(R_{abcd} - \lambda(g_{ac}g_{bd} + g_{ad}g_{bc}))V^{cd} = 0,$$

where $V^{cd}$ is an anti-symmetric tensor. The requirement ensures that solutions $\lambda$, whose existence we assume, are independent real scalar functions. While Komar takes these scalars to be preferred, since "they are the only nontrivial scalars which are of least possible order in derivatives of the metric"[72], p. 1183, we do not need to take the simplest choices. Nonetheless, to keep our notation tractable, we will use sets that have the next-to-lowest order of derivatives (namely, one in which we take the D'Alembertian of one curvature scalar). Suppose that we choose

$$R_{(A)} = (\text{Riem}^2 - \text{Weyl}^2, \Box R, \text{Ric}^2, \Box \text{Weyl}^2) \qquad (18)$$

as a concrete candidate for the $\chi$-fields. Here, $\text{Ric}_{ab} = R^c{}_{acb}$ denotes the Ricci curvature tensor, $R = g^{ab}\text{Ric}_{ab}$ the scalar curvature, and Weyl the Weyl curvature tensor. The reason we take $\Box R$, and not $R$, and similarly with $\text{Weyl}^2$, is so that we do not have to keep track of the following functional dependence relation: $\text{Weyl}^2 = \text{Riem}^2 - 2\text{Ric}^2 + \frac{1}{3}R^2$. With this choice, a particular point $p \in \mathcal{M}_1$ is defined as the point at which the curvature scalars take on specific values, for instance

$$(\text{Riem}^2 - \text{Weyl}^2)^{(1)}(p) = 1, \; (\Box R)^{(1)}(p) = 2, \; (\text{Ric}^2)^{(1)}(p)$$
$$= 3, \; (\Box \text{Weyl}^2)^{(1)}(p) = 4. \qquad (19)$$

This point will have a counterpart with respect to the comparison map defined by the curvature scalars $R_{(A)}$ in $\mathcal{M}_2$, namely the

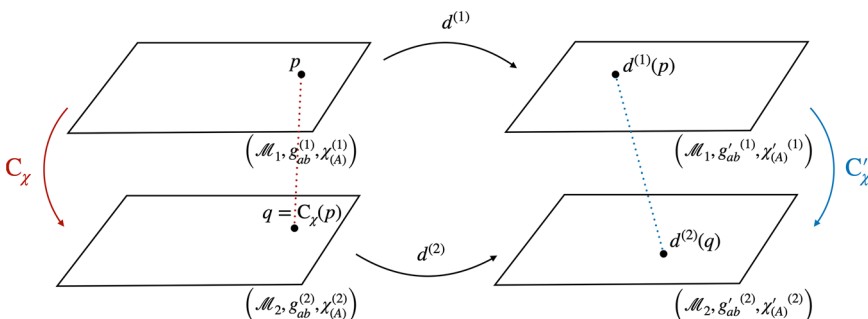

**Fig. 12 | The first step of a QRF transformation: the application of a quantum diffeomorphism.** The quantum diffeomorphism acts on a superposition of models $(\mathcal{M}_1, g^{(1)}_{ab}, \chi^{(1)}_{(A)})$ and $(\mathcal{M}_2, g^{(2)}_{ab}, \chi^{(2)}_{(A)})$ with the comparison map $C_\chi$ established through the $\chi$-fields. In particular, with respect to $C_\chi$, the point $p \in \mathcal{M}_1$ is identified with the point $q = C_\chi(p) \in \mathcal{M}_2$. We say that the pair $(p, q)$ is *localised* with respect to the $\chi$-fields. The quantum diffeomorphism consists of $d^{(1)}$ and $d^{(2)}$ in the two branches of the superposition and maps to the models $(\mathcal{M}_1, g'^{(1)}_{ab}, \chi'^{(1)}_{(A)})$ and $(\mathcal{M}_2, g'^{(2)}_{ab}, \chi'^{(2)}_{(A)})$, respectively. It changes the comparison map $C_\chi$ to $C'_\chi$ in such a way that the pair $(p, q)$ remains localised relative to the new comparison map $C'_\chi$, specifically, $d^{(2)}(q) = C'_\chi(d^{(1)}(p))$. In this sense, the quantum diffeomorphism leaves the identification of points intact. Note that a full QRF transformation involves a second step, not depicted in this figure, namely a change of section, i.e. a change to the corresponding quantum reference fields $\tilde{\chi}$. This step changes the comparison map to $C_{\tilde{\chi}}$, which will, in general, no longer identify $d^{(1)}(p)$ and $d^{(2)}(q)$: $d^{(2)}(q) \neq C_{\tilde{\chi}}(d^{(1)}(p))$.

point $q \in \mathcal{M}_2$ at which

$$(\text{Riem}^2 - \text{Weyl}^2)^{(2)}(q) = 1, \ (\Box R)^{(2)}(q) = 2, \ (\text{Ric}^2)^{(2)}(q)$$
$$= 3, \ (\Box \text{Weyl}^2)^{(2)}(q) = 4. \tag{20}$$

But, of course, we could have also chosen a different set of scalars as quantum reference fields, for example

$$\tilde{R}_{(A)} = (\text{Riem}^2, \Box R, \text{Ric}^2, \Box \text{Weyl}^2), \tag{21}$$

These two choices of scalar fields are related by a QRF transformation. More specifically, there exists a quantum diffeomorphism consisting of $d^{(1)}$ and $d^{(2)}$ and a change of section which takes us from the reference frame of $R_{(A)}$ to the reference frame of $\tilde{R}_{(A)}$.

Now, just because one set of scalar fields takes on the same values at $p \in \mathcal{M}_1$ and $q \in \mathcal{M}_2$, this does not have to be the case for the second set of scalar fields. In particular, the two might be in superposition with respect to each other. For example, we might have $(\text{Riem}^2)^{(1)}(p) = 1$ and $(\text{Riem}^2)^{(2)}(q) = 2$, while $(\text{Weyl}^2)^{(1)}(p) = 0$ and $(\text{Weyl}^2)^{(2)}(q) = 1$. These properties are not changed by the quantum diffeomorphism, since $(d_*^{-1} \tilde{R}_{(A)})(d(p)) = \tilde{R}_{(A)}(p)$ for any diffeomorphism $d$. As a result, when comparing with respect to the $\tilde{R}_{(A)}$-fields, the point $d^{(1)}(p) \in \mathcal{M}_1$ will *not* be identified with the point $d^{(2)}(q) \in \mathcal{M}_2$ but rather with the, in general inequivalent, point $q' \in \mathcal{M}_2$ at which the new reference fields take on the same specific set of values. As a result, the pair $(d^{(1)}(p), d^{(2)}(q))$ will no longer be localised with respect to the $\tilde{R}_{(A)}$-fields.

Thus, using point coincidences to identify points on spacetimes across a superposition—namely, identifying those points to which the same set of field values are associated—we remove the redundancy of the diffeomorphism invariance; however, there remains the ambiguity of *which* quantum reference fields to use to establish this identification. As a response, we should be careful not to assign (too much) physical meaning to the identification and localisation of spacetime points — even when characterised in terms of physical fields. This, we consider to be the main consequence of the *quantum hole argument* beyond the implications of its classical counterpart.

**Localisation of events, interference, and recombination**. In particular, a QRF transformation also changes the localisation of physical events as defined, for example, through worldline crossings. Suppose that you are interested in not just any pair $(p, q)$ of spacetime points but in the points associated with a given physical event, such as the interaction between two particles. Let us further assume that the interaction occurs when the worldlines of the two particles cross and that this happens at a localised pair of points relative to the original QRF. That is, consider a situation in which the worldlines of the particles cross at a point $p \in \mathcal{M}_1$ and at point $q = C_\chi(p) \in \mathcal{M}_2$. After performing a QRF transformation, the crossing of the particles' worldlines will no longer be associated with the pair $(p, q)$. This is because the quantum diffeomorphism in the first step (i) of the QRF transformation drags along the particles' worldlines. As a result, they will cross at the points $d^{(1)}(p)$ and $d^{(2)}(q)$ in the different branches of the superposition respectively. Moreover, as we have seen, the second step of the QRF transformation changes the comparison map such that, in general, $d^{(2)}(q) \neq C_{\tilde{\chi}}(d^{(1)}(p))$, implying that the pair of points is delocalised in the new frame of reference.

At first sight, this may raise worries when considering an interaction that is defined to take place at a localised event, since that event can be mapped to a delocalised one through a change of QRF. Thus, Adlam et al. have argued: "it would seem that using different identities across branches would produce different predictions, since we would end up with different combinations of fields interacting"[66], p. 13, cf. also pp. 33–36. However, while a QRF transformation may change the localisation of an event at which the interaction is taking place, it does not change the locality of the interaction itself. The latter is a notion defined separately in each branch. It simply refers to whether the fields interact at a single point *within* a given

spacetime, not whether it is the same point across the different branches. Therefore, the interaction of the fields remains the same in each branch, with no empirical consequences upon changes of QRFs. While this only concerns the interaction *within* a given spacetime, Adlam et al. are particularly concerned with phenomena occurring *across* branches, such as interference or recombination:

"Consider for illustration the BMV experiment [our refs. [78],[79]] in which two masses in superposition states are taken to get entangled with each other through gravitational interaction (and gravitational interaction alone): the relevant literature tacitly assumes that the location of the masses are all relative to one joint lab frame–no matter whether the experiment is modelled through a Newtonian potential, or (low-energy) metric fields, as done by Christodoulou and Rovelli (2019) [our ref. [68]]. But if we perform a quantum diffeomorphism which shifts the point at which the recombination occurs within one of the branches, then the phase change will be different and the interference effects will change."[66], p. 13

We agree "that the location of the masses are all relative to one joint lab frame"—in fact, the quantity that is recombined (i.e. becomes the same across the branches) is precisely the *relative* location of each mass with respect to the lab frame, a manifestly frame-independent quantity. It is only in the reference frame of the laboratory itself that this relative quantity, $x_M - x_{lab}$, is equal to the coordinate position $x_M$ of the mass. One is of course free to describe the BMV setup in a different QRF; however, the location of each mass becoming definite across the branches no longer indicates that the mass gets recombined with respect to the experimental setup. Agreed, the recombination can now occur in a superposition of locations relative to the new frame. Nevertheless, we see no reason why the outcome of the BMV experiment would be changed under a QRF transformation, because any operations that one might think depend on the coordinate position of the mass, in fact depend on its relative location with respect to the lab frame. For example, let us consider a setup in which each mass is placed in a superposition and gets recombined using beamsplitters. In this case, for each mass, the phase difference is calculated as the difference in phase accumulated along its path in one and the other branch, where the start and end points of the paths are given by the coincidence of the mass with the first and second beamsplitter, respectively. Only in the laboratory reference frame, in which the beamsplitters are localised across the branches, are these coincidences also localised. Note that if one wanted to model explicitly the recombination of the spacetimes sourced by the masses in the BMV setup in a general relativistic language, one would first have to introduce a $(3 + 1)$-split that enables one to consider the recombination of three-dimensional hypersurfaces in some chosen time direction. However, treating such cases explicitly goes beyond the scope of the present paper.

**Relational observables**. To get a more encompassing understanding of the effects of changing the QRF and the identification of points, let us further consider its implications for observables. Let us denote a partial observable[82] on a classical spacetime by a function $\mathcal{O}[g_{ab}, \Psi_{matter}] : \mathcal{M} \to \mathcal{S}$ that depends on the metric and matter fields as well as arbitrary derivatives thereof. For simplicity, we restrict ourselves in the following discussion to scalar functions for which $\mathcal{S} = \mathbb{R}$. On a superposition of spacetimes, a partial observable is then given by a collection of classical partial observables, $\mathbf{O} := \{\mathcal{O}^{(i)} \equiv \mathcal{O}[g_{ab}^{(i)}, \Psi_{matter}^{(i)}]\}_i$ where $i$ denotes the respective branch of the superposition. Given a partial observable and a choice of reference frame $\chi = \chi_*$ with corresponding section $\sigma$, one can define the *relational* or *dressed observable* $\mathbf{O}_{|\chi=\chi_*} := \{\mathcal{O}^{(i)}_{|\chi=\chi_*}\}_i := \{\mathcal{O}^{(i)} \circ (\chi^{(i)})^{-1}|_{\chi^{(i)}=\chi_*}\}_i$. Note that, when using idealised reference fields (in the sense of the first of the three options listed at the start of the "Quantum coordinate fields and quantum reference frame changes" subsection), these relational observables can still capture any

possible configuration (state) of the metric relative to these reference fields. When using dynamically coupled reference fields (in the sense of either the second or the third of the three options above), on the other hand, fixing the values of the reference fields also leads to a unique gauge fixing for the spacetime geometry. This distinction is discussed further in ref. 83. $O_{|\chi=\chi_*}$ is a set of functions from $\mathbb{R}^4$ (understood as the range of the four $\chi$-fields, taken together), back to the respective spacetime manifolds $\mathcal{M}_i$, and then to the value-space $\mathcal{S}$, where the $i$th function gives the value of the partial observable $\mathcal{O}^{(i)}$ when the reference field $\chi^{(i)}$ takes the value $\chi_*$. These relational or dressed observables have not only been studied in the context of QRFs[7,14,74] but also play an important role in certain approaches to quantum gravity[75,82,84–86]. An example of a partial observable would be the curvature scalar **R**, which takes a point in $\mathcal{M}_i$ and returns the value of the curvature scalar for the metric $g_{ab}^{(i)}$ on each manifold in the superposition. When choosing a reference frame $\chi = \chi_*$, we get the corresponding relational observable $\mathbf{R}_{|\chi=\chi_*}$ which is a set of maps from the space $\mathbb{R}^4$ of values for the $\chi$-fields to $\mathcal{S} = \mathbb{R}$. Here is a concrete example: given a value for the scalar fields, $x \in \mathbb{R}^4$, it returns the curvature scalar $R^{(i)}(p_i) = R^{(i)}((\chi_*^{(i)})^{-1}(x))$ for $p_i = ((\chi_*^{(i)})^{-1})(x)$ on each manifold in the superposition.

We can now ask ourselves the question of whether a given relational observable is in a superposition, i.e. takes different values across the branches of a coherent superposition, in a given reference frame. To see this, take again a pair of points $(p, q)$ as above, identified via the comparison map $C_\chi$ established by the reference fields $\chi$. To check whether a given partial observable **O** is in superposition or "definite" at $(p, q)$, we compare the value that the partial observable $\mathcal{O}^{(1)}$ takes at $p \in \mathcal{M}_1$ with the corresponding value that $\mathcal{O}^{(2)}$ takes at $q = C_\chi(p) \in \mathcal{M}_2$. If the two values coincide, that is, $\mathcal{O}^{(2)}(q) = \mathcal{O}^{(1)}(p)$, we say that the observable is definite, i.e. not in superposition. If this is not the case, we can conclude that the observable is in a superposition at $(p, q)$. Importantly, this relation is invariant under the action of any quantum diffeomorphism. To see this, take, as above, a quantum diffeomorphism consisting of $d^{(1)}$ and $d^{(2)}$. Then, whenever $\mathcal{O}^{(2)}(q) = \mathcal{O}^{(1)}(p)$, we also have

$$\begin{aligned}\mathcal{O}'^{(2)}(d^{(2)}(q)) &= (\mathcal{O}^{(2)} \circ (d^{(2)})^{-1} \circ d^{(2)})(q) = \mathcal{O}^{(1)}(p) \\ &= (\mathcal{O}^{(1)} \circ (d^{(1)})^{-1} \circ d^{(1)})(p) = \mathcal{O}'^{(1)}(d^{(1)}(p)).\end{aligned} \tag{22}$$

Thus, just as no quantum diffeomorphism can change the localisation of physical events, it also cannot change whether an observable is in superposition or definite. If, however, we change the reference frame by choosing a different set of fields $\tilde{\chi}$, generally one will find $\mathcal{O}'^{(2)}(q') \neq \mathcal{O}'^{(1)}(d^{(1)}(p))$ with $q' = C_{\tilde{\chi}}(d^{(1)}(p))$.

Thus, a QRF transformation can change whether an observable appears to be in superposition. Note, however, that this should not come as a surprise; after all, the relational observable $\mathbf{O}_{|\chi=\chi_*}$ describes a different quantity from $\mathbf{O}_{|\tilde{\chi}=\tilde{\chi}_*}$. Referring back to the translation example of a massive object in superposition, this is evident: the relative distance of the particle $P$ to the massive object $M$ and the relative distance of $P$ to some other system represent different physical quantities. Similarly, the curvature relative to the $\chi$-fields is a different quantity than the curvature relative to the $\tilde{\chi}$-fields.

**Relation to indefinite causal order.** The localisation of events plays a crucial role in the study of indefinite causal order — a phenomenon that occurs in processes in which events can no longer be assigned a classical causal ordering. In a recent paper[19] some of us have studied the effect of quantum diffeomorphisms on the localisation of events. With the refined understanding of localisation and identification gained through the present work, we can revisit these findings and view them in a new light.

Let us briefly introduce the problem. One prime example of a process with indefinite causal order is the *quantum switch*[42,43], in which two agents apply operations on a target system in a superposition of opposite orders. The application of each operation marks an event, often represented as the

target system passing through the laboratory of the respective agent. There are different implementations of the quantum switch. In the *optical quantum switch*[44,87], the indefinite causal order arises through a superposition of paths of the target system on a definite Minkowski spacetime. In the *gravitational quantum switch*[88], it is realised through a gravitational field in superposition. When sketching both processes in a spacetime diagram, the optical switch is depicted as a "four-point" switch in which the target system passes through the laboratories at four different spacetime points while the gravitational implementation can be constructed as a "two-point" switch[46].

There is an ongoing debate on whether a proper realisation of indefinite causal order requires a superposition of spacetimes or whether an implementation in Minkowski spacetime, such as the optical switch, is sufficient[44–47,89,90]. A crucial question in this debate is how many events there are in the different implementations of the quantum switch[46,47,89,91,92]. The answer to this question of course depends largely on how one defines a quantum event. In ref. 19, some of us proposed a notion of event based on worldline coincidences which encompasses situations where both the trajectories and the spacetimes can be in superposition. Given a superposition of models $\varphi_i$, $i = 1, 2$, each encoding the configuration of the spacetime metric as well as several timelike worldlines, we defined an event by the pair of points $p \in \mathcal{M}_1$ and $q \in \mathcal{M}_2$ at which the worldlines cross in the respective manifolds. To relate the different implementations of the quantum switch, we then applied a quantum-controlled transformation that maps a four-point to a two-point switch, thereby localising the events.

Note that in ref. 19 we referred to these transformations as quantum diffeomorphisms while using the identity map to compare points across the superposition both before and after the transformation. In light of the understanding developed in this article, we can view this as a QRF transformation. As we have seen above, such a transformation can indeed alter the localisation of events. In a reference frame in which an event is localised, we can denote its location by the same manifold point. However, in a different QRF in which it is delocalised, we would have to use different manifold points to describe its location. Thus, under a QRF transformation, the number of "different" manifold points associated to a given event can change. Therefore, it would not be physically meaningful to take the number of different manifold points as the number of events, as has been suggested in ref. 46. In other words, we would argue against taking the spatiotemporal location as an inherent property of an event, as this is not invariant under QRF changes.

## Conclusions

This paper has had two main goals. The first goal was to develop a framework for quantum reference frames that places the focus on their conceptual aspects and allows for a clear visualisation of the main features of a QRF transformation. We developed such a framework by using the ideas of refs. 25,26,30–32 on symmetries and representational conventions and extending them to the quantum level. This understanding of QRFs was illustrated using several different examples, such as a simple translation-invariant setup, general QRF transformations for locally compact groups, and QRF transformations for the diffeomorphism group. Moreover, we explained how the present framework can accommodate even "incomplete" reference frames, which carry a non-trivial stabiliser group[14]. The second goal was to apply this framework to analyse its conceptual implications for scenarios involving semi-classical spacetimes in superposition, specifically the consequences of QRF transformations for the identification and localisation of events and relational observables, resulting in the formulation of the quantum hole argument.

Accordingly, we will now summarise the work done towards these two goals; and then we will close by briefly discussing the relation of this paper to other approaches to QRFs and related fields, as well as open questions and directions for future research.

The understanding of QRF transformations, which emerge from this work, can be summarised as follows. A QRF transformation is essentially composed of two steps: (i) a quantum-controlled symmetry transformation, and (ii) a change in how we identify configurations or locations across

different branches in a superposition. In the space of models, the first step is realised by moving the models $\varphi$ along their respective orbits using different group elements $g_\sigma(\varphi)$. The second step corresponds to a change in the choice of section $\sigma$ in the space of models and thus a change in the counterpart relation $\text{Counter}_\sigma(\varphi, \varphi')$, which prescribes how to compare models in different orbits. This step is necessary because, in the presence of symmetries, there is no preferred way of comparing different possible configurations of a system. Rather, it is the choice of the section on the classical level and the choice of QRF at the quantum level that picks out a preferred means of identification via the counterpart relation. When considering a superposition of different general-relativistic spacetimes, it implies that whether two spacetime points are said to be "the same" or "different" across the superposition, depends on the choice of QRF.

Both of these steps are crucial to the conceptual understanding of some of the key features of QRF transformations. Only if we change the configurations through a quantum-controlled symmetry transformation *and* change our way of identifying configurations or locations across the branches of a superposition, can we ensure that the state of the reference frame factorises out—i.e. is "the same" across all branches—and alter whether a system is in a superposition—that is, in "different" configurations across the branches. Once one understands that the way we compare objects across a superposition is not pre-defined but depends crucially on the choice of QRF, the frame-dependence of both superposition and entanglement finds a natural explanation. While the relativity of sameness and difference can already be understood at the classical level when comparing different possible configurations, it has more physical implications when one considers quantum superpositions. Similarly, in the case of ideal QRFs, the relativity of subsystems[2,10,13,14,23] can, maybe surprisingly, already be understood classically through a reshuffling of the degrees of freedom (cf. ref. 23), but becomes particularly relevant when studying entanglement in quantum theory. Note, however, that it is still an open question whether subsystem relativity for non-ideal frames, that is, frames that do not carry a representation in $L^2(G)$[10], can be captured by this explanation.

In the subsequent sections, we focused on scenarios involving semi-classical spacetimes in superposition. There, we considered QRF transformations for the diffeomorphism group, which can be understood as changes of *internal* or *physical* reference fields $\chi$. By using these fields to identify points across the different spacetimes in superposition, we provided not only a physically intuitive implementation of the counterpart relation, but also a tool to investigate in detail how the localisation of spacetime points, events, and fields changes under a QRF transformation. It became clear that, even when identifying points through coincidences of physical field values, and thus removing the redundancy of the diffeomorphism invariance, there remains an ambiguity in choosing *which* quantum reference fields to use to construct this identification. Thus, a change of QRF can alter the localisation of spacetime points, events, and fields. As a result, we argued that an invariance under QRF transformations gives rise to a stronger version of the hole argument, calling into question the metaphysics of not just spacetime points but also their identification across manifolds in superposition. We discussed the implications of the change in localisation of events under a QRF transformation for the interaction of fields as well as interference experiments, in particular the BMV setup. We also illustrated how QRF transformations can change whether an observable appears in a superposition or not. Finally, we discussed the implications of the change in localisation of events under a QRF transformation for the study of indefinite causal order, arguing that spacetime localisation should not be regarded as an inherent property of a quantum event and that therefore both the optical and gravitational switch can be regarded as two-event switches.

We close by briefly discussing the relation of this paper to other approaches to QRFs and related fields and the open questions and directions for future research.

One of the most pressing problems is, of course, the rigorous formalisation of both the measures over the diffeomorphism group and the space of metrics as well as the construction of a well-defined Hilbert space for the spacetime metric. While some important advances have been made

in this regard, e.g. in the context of lattice and algebraic quantum field theory or spin foams[93–95], these problems have remained unsolved for several decades. We believe that this should not hinder our efforts to understand some of the novel features of non-classical spacetimes with the tools that we do have on hand. It is true, of course, that we cannot exclude the possibility that a rigorous solution to the aforementioned problems might reveal unforeseen effects and it is therefore important to find a balance between addressing the known challenges and making progress with current techniques. Moreover, advancements in both areas can complement each other and thus guide the development of a more complete theory in the future.

Moreover, while our framework is mainly based on the understanding of QRFs as in refs. 2,5,8,11,17,18, it can be connected to many other approaches towards making physics relational. Firstly, visualising QRF transformations in the space of models makes clear that they have a classical analogue—the model- or configuration-dependent symmetry transformations. When considering field theories, these become *field-dependent* maps, such as the changes between dynamical reference frames in refs. 74,96,97. These, in turn, are directly related to edge modes, additional degrees of freedom that arise when considering a bounded region of spacetime in gauge theory and gravity (e.g. refs. 98–100). As shown in refs. 21,101–103, edge modes can be understood as dynamical reference frames and used to construct relational observables. On the quantum level, they can be seen as QRFs and define a relational evolution of the quantum state for generally covariant theories. When constructing relational observables, both edge modes and dynamical reference frames act as dressing functions that turn the bare observables into gauge-invariant quantities relative to the reference fields[102]. This brings us back to the space of models, where the group elements $g_\sigma$ that map the models to a chosen section $\sigma$ can equally be seen as dressings for the models[26].

Secondly, while the connection of the present framework to QRF approaches based on observable algebras[1,22] as well as the framework introduced in ref. 13 remains an open question, it is possible to connect it to some aspects of the perspective-neutral approach to QRFs[3,7,14,23]. It has been shown[14] that the latter is equivalent to the approach of[2,8] for ideal QRFs, so our framework carries over directly in this context. Moreover, as discussed in the "Locally compact groups and imperfect quantum reference frames" subsection, the present work encompasses non-ideal QRFs with non-trivial stabiliser groups, as developed in ref. 14. What is still to be worked out is the formal treatment of non-ideal QRFs using coherent states. Finally, recent advances in the study of gravitational observable algebras[104–112] have raised important questions about their connection to quantum reference frames[49,50,113–115].

Note further, that the perspective-neutral approach is formulated in the phase space picture, where the physical states live in the zero-charge sector. Our space of models, on the other hand, solely represents configuration space and therefore makes no restrictions on the charge. It would, however, be interesting to understand how the different charge sectors could be represented within our framework. This would require considering the entire phase space rather than focusing on configuration space; here the work of Gomes and Riello[96,116,117] on the infinitesimal version of the counterpart relation would likely prove to be useful. This could also provide the connection to Noether charges and superselection sectors and thereby bring us back to the motivation of early works on QRFs[24]. Moreover, considering superpositions of states in different charge sectors could lead to QRF transformations which are characterised by a branch-wise phase change in addition to the quantum-controlled symmetry transformation considered in the present work. Exploring the consequences of such modified QRF transformations on quantum states and observables will be the focus of future work.

Finally, the present work prompts several philosophical questions. To what extent do the present results and, more generally, our understanding of QRFs depend on any particular interpretation of quantum theory? What meaning would different interpretations assign to the quantum state relative to a particular QRF as well as to the comparison map? Does the quantum hole argument formulated as above challenge a realist interpretation of

quantum mechanics? More broadly, considering the growing number of recent insights regarding the relational nature of concepts, previously thought to be absolute—such as superposition, entanglement, subsystem structure, localisation of events, and sameness or difference—in light of *quantum* symmetries and reference frames, one starts to wonder whether we need to let go of the absoluteness of further notions in our understanding of the physical world.

*Note added*: At around the same time of this article's first appearance as a pre-print, a paper[118] came out with similar conclusions about the relative localisation of events in the context of linearised general relativity and Jackiw–Teitelboim gravity. The overlap between these approaches clearly deserves further study.

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

## Acknowledgements

We thank Emily Adlam, Marios Christodoulou, Lucien Hardy, Philipp Höhn, Niels Linnemann, James Read, Robin Simmons, Daniel Vanzella, and V. Vilasini for stimulating discussions and helpful comments on an earlier version of this work. This research was funded in whole or in part by the Austrian Science Fund (FWF) [10.55776/F71] and [10.55776/COE1]. For open access purposes, the author has applied a CC BY public copyright license to any author-accepted manuscript version arising from this submission. Funded by the European Union—NextGenerationEU. V.K. acknowledges support through a DOC Fellowship of the Austrian Academy of Sciences as well as a grant from the Blaumann Foundation. H.G. would like to thank the British Academy. J.B. and H.G. would like to thank Oriel College Oxford and Trinity College Cambridge, respectively. They both thank the Austrian Academy of Science (ÖAW), IQOQI Vienna, and the QISS research network for support and for very generous hospitality in Vienna. This publication was made possible through the financial support of the ID 61466 and ID 62312 grants from the John Templeton Foundation, as part of The Quantum Information Structure of Spacetime (QISS) Project (qiss.fr). The opinions expressed in this publication are those of the authors and do not necessarily reflect the views of the John Templeton Foundation.

## Author contributions

V.K., A.-C.d.l.H., L.A., C.C., H.G., J.B. and Č.B. contributed to all aspects of the research, with leading and equal input of V.K., A.-C.d.l.H. and L.A.

## Competing interests

The authors declare no competing interests.
