## [Transparent Peer Review file · Communications Physics]

Quantum Reference Frames, Localisation of Events, and the Quantum Hole Argument

Corresponding Author: Ms Viktoria Kabel

Version 0:

Reviewer comments:

Reviewer #1

(Remarks to the Author)

Since I have already given a detailed review of this manuscript, I will keep this one short. Let me first reiterate that I think this is a good paper, is nicely written, largely clear, and deserves to be published. I think it deserves to be published in this venue.

The authors have made a decent effort to engage with my previous criticisms, and I thank them for the well organised and detailed response. I still have a couple of lingering bits of unease, detailed below, which the authors might like to chew on a bit before sending their final version, but basically I think this paper is ready for publication as is.

A. Not important: I think "section" in mathematics in general does not require anything beyond beyond "right inverse". See e.g. Lawvere.

B. 4. RC: I'm not sure what is meant by "express...observables in these other bases".

AR: The symmetry group of the theory induces a preferred basis.

RC': Not necessarily, precisely speaking.

AR: For any choice of basis of the Hilbert space, there exists a decomposition.

RC': Of what?

AR: Observables can be decomposed into several bases, including their eigenbasis

(spectral decomposition), the symmetry group basis, or the "momentum basis" (conjugate basis). Thus by expressing observables in these other bases, we mean decomposing them in other bases. We modified this on p. 6.

RC': I just don't think this is true in general, without some quite good theorems. How to I write the position in the number basis?

C. 5: Ok, but I think the Bargmann SSR is quite different from the photon number "SSR", but this is beyond the scope I think!

Reviewer #2

(Remarks to the Author)

The paper is focused on applying the philosophical perspective and language of Gomes and Butterfield, Refs.~[25,26,30-32] to an interpretation of changes of quantum reference frames. This manifests through defining a counterpart map in Eq.~(2) that allows the authors to compare models that may or may not lie on the orbit of a symmetry group present in the considered theory. The authors then examine a few examples and comment on conceptual implications for the hole argument, BMV-type experiments, relational observables, and indefinite causal order.

The paper reads more like a review article, primarily focused on synthesizing the authors' past contributions. The novel content appears to be imbuing the description with the philosophical terms: (1) space of models, (2) representational convention, and (3) counterpart relation; as well as the discussion of several conceptual issues.

The paper is well-written and pedagogical, as the authors claim. It will interest those working on quantum reference frames, quantum foundations, and the philosophy of physics, and the work constitutes an advance in the field through the connections made between QRF and Refs.~[25,26,30-32]. Provided the comments below are taken into account, I believe this paper meets the criteria for publication in Communication Physics.

Comments:

1. At the end of the fourth page, the claim about global sections in gauge theory is misleading. Part of the interest in gauge theories is the non-trivial topology of the fiber bundle, which is eliminated by the presence of a trivializing global section. The authors should qualify this statement.
2. The Gribov ambiguity has been analyzed in the QRF context by 1809.05093, 2405.13884, and 2410.11029. The presence of inequivalent QRFs is akin to having different superselection sectors, which is analogous to having non-overlapping gauge fixings. The authors should include this in a more extended discussion.
3. In the discussion of a symmetry group G , the authors claim that this induces a preferred basis for models. On what grounds is this basis preferred? It may be natural mathematically but I believe the statement has to be qualified as it is as good as any other basis as the authors remark in the remainder of the relevant paragraph.
4. It is important to note that frame dependence is not a quantum phenomenon. The authors point to superposition and entanglement as being frame-dependent but those are only features of quantum theory which are frame-dependent. This would be good to emphasize at the end of Section II.
5. It might be important to point out that since the authors employ different group elements in different orbits, this leads to the possibility (as in the Gribov problem) to the degeneration of the constraint group. In other words, the theory could break up into different groups H, G, K, \dots which gives the model space the complexion of an orbifold.
6. In the second-to-last paragraph of the tenth page, the authors refer to an example of a tripartite system. This was worked out in their Ref. [10] but is not referenced there.
7. In Section IV, the authors refer to locally compact groups but then mention quantum diffeomorphisms at the end. Do these constitute a locally compact group?
8. In the second to last paragraph of Sec. IV, the authors discuss the possibility of using coherent states with respect to the group of reference frame orientations, which may not be orthogonal. The authors claim that such discussions have not appeared in the context of QRF they considered and only cite one of the author's works. Yet such coherent state systems were introduced in this context for temporal reference frames in 1712.00081, and further studied in 1912.00033 and 2007.00580, after which Ref. [14] appeared, which is an extension to unimodular Lie groups built on these results. These works discuss changes of QRFs that are equivalent to what the authors discuss, so it is not clear what the authors mean when they claim "This extension to coherent state systems has not been implemented in the QRF approach of [2,5,8,11,17,18]". The authors should clarify this point and cite past works where appropriate.
9. In the main example of the manuscript, the authors consider a superposition of semiclassical spacetimes. This is a general point of frustration in the literature, which becomes a particular point for the authors, but the description of such objects is still lacking. What does it mean to have a linear Hilbert space for quantum gravity? One can imagine a state with support on multiple backgrounds with overlaps computed using a path integral for instance. However, the description provided here is vague and would be improved if made quantitative.
10. The references on the crossed product in quantum gravitational algebras and their relation to QRFs are incomplete. The following references are relevant: 2112.12828, 2206.10780, 2306.07323, 2312.16678, 2405.13884, 2407.01695, 2410.11029, 2207.06704, 2302.14747, 2304.11845, and 2303.02837.

Reviewer #3

(Remarks to the Author)

I co-reviewed this manuscript with one of the reviewers who provided the listed reports. This is part of the Communications Physics initiative to facilitate training in peer review and to provide appropriate recognition for Early Career Researchers who co-review manuscripts.

Version 1:

Reviewer comments:

Reviewer #2

(Remarks to the Author)

I have reviewed the authors replies to my comments and am happy with their responses and updates to the paper. I thus recommend the paper for publication.

Reviewer #3

(Remarks to the Author)

I co-reviewed this manuscript with one of the reviewers who provided the listed reports. This is part of the Communications Physics initiative to facilitate training in peer review and to provide appropriate recognition for Early Career Researchers

who co-review manuscripts.

Author Response to Reviews of

Identification is Pointless: Quantum Reference Frames, Localisation of Events, and the Quantum Hole Argument

V. Kabel, A.-C. de la Hamette, L. Apadula, C. Cepollaro, H. Gomes, J. Butterfield, Č. Brukner
Manuscript ID: NCOMMS-24-22993

Response to Reviewer 1

RC: *Since I have already given a detailed review of this manuscript, I will keep this one short. Let me first reiterate that I think this is a good paper, is nicely written, largely clear, and deserves to be published. I think it deserves to be published in this venue. The authors have made a decent effort to engage with my previous criticisms, and I thank them for the well organised and detailed response. I still have a couple of lingering bits of unease, detailed below, which the authors might like to chew on a bit before sending their final version, but basically I think this paper is ready for publication as is.*

AR: We thank the referee for studying our manuscript and replies so carefully and for suggesting our paper for publication. Below, we have addressed their remaining comments individually. All new changes in the manuscript are highlighted in teal, while changes implemented during the first round of the review are marked in blue for clarity.

RC: *A. Not important: I think "section" in mathematics in general does not require anything beyond beyond "right inverse". See e.g. Lawvere.*

AR: Judging from the reference, we assume that the referee is referring to sections in the category theory sense. When speaking about sections in our manuscript, we generally mean a section of a fiber bundle (a smooth, or at least continuous, right inverse of the projection function).

RC: *B. 4. RC: I'm not sure what is meant by "express...observables in these other bases".*

AR: *The symmetry group of the theory induces a preferred basis.*

RC': *Not necessarily, precisely speaking.*

AR: *For any choice of basis of the Hilbert space, there exists a decomposition.*

RC': *Of what?*

AR: *Observables can be decomposed into several bases, including their eigenbasis (spectral decomposition), the symmetry group basis, or the "momentum basis" (conjugate basis). Thus by expressing observables in these other bases, we mean decomposing them in other bases. We modified this on p. 6.*

RC': *I just don't think this is true in general, without some quite good theorems. How to I write the position in the number basis?*

AR: We agree that in general there are subtleties one needs to be careful about, especially with infinite dimensional Hilbert spaces/field theoretic models. In the case of standard quantum mechanical systems, including the specific example brought up by the referee, this is relatively straightforward, though. For example, for the harmonic oscillator, there is a closed form expression for the position-number transition amplitudes, namely $\langle x|n\rangle = \frac{1}{\sqrt{2^n n!}} \pi^{-1/4} \exp(-x^2/2) H_n(x)$, where $H_n(x)$ are the Hermite polynomials.

We have slightly modified the relevant statement in the article on p. 6 to now read "*But of course one can*

express states and observables in an arbitrary basis. [Footnote: Here, we set aside subtleties that may arise in the context of field theoretic models and in particular in the context of spacetimes in superposition that will be discussed in Sec. V.] In particular, in the translation group example described in the next Section, the preferred basis is the position basis but one could straightforwardly express the states and observables in other bases as well."

RC: *C. 5: Ok, but I think the Bargmann SSR is quite different from the photon number "SSR", but this is beyond the scope I think!*

AR: While we appreciate the referee's point about the subtle difference between the photon number SSR and the Bargmann SSR, we believe that a detailed discussion of various superselection rules is beyond the paper's scope.

AR: We thank the referee once again for assessing our work and our response to the previous round of comments. We hope this reply addresses and resolves any remaining points.

Response to Reviewer 2

RC: *The paper is focused on applying the philosophical perspective and language of Gomes and Butterfield, Refs. [25,26,30-32] to an interpretation of changes of quantum reference frames. This manifests through defining a counterpart map in Eq. (2) that allows the authors to compare models that may or may not lie on the orbit of a symmetry group present in the considered theory. The authors then examine a few examples and comment on conceptual implications for the hole argument, BMV-type experiments, relational observables, and indefinite causal order.*

The paper reads more like a review article, primarily focused on synthesizing the authors' past contributions. The novel content appears to be imbuing the description with the philosophical terms: (1) space of models, (2) representational convention, and (3) counterpart relation; as well as the discussion of several conceptual issues.

The paper is well-written and pedagogical, as the authors claim. It will interest those working on quantum reference frames, quantum foundations, and the philosophy of physics, and the work constitutes an advance in the field through the connections made between QRF and Refs. [25,26,30-32]. Provided the comments below are taken into account, I believe this paper meets the criteria for publication in Communication Physics.

AR: We thank the referee for studying our manuscript and replies so carefully. Below, we have addressed their remaining comments individually. All new changes in the manuscript are highlighted in teal, while changes implemented during the first round of the review are marked in blue for clarity.

RC: *1. At the end of the fourth page, the claim about global sections in gauge theory is misleading. Part of the interest in gauge theories is the non-trivial topology of the fiber bundle, which is eliminated by the presence of a trivializing global section. The authors should qualify this statement.*

AR: While we agree that in gauge theories, in particular for non-abelian gauge groups, one has to be quite careful with the definition of sections due to the non-triviality of fibre bundles, this does not hold true in general. For example, in electrodynamics a global section can be found, which is why we wrote “(global) section” in brackets. To make this clearer, we replaced “(global)” with “(not in general global)” on p. 4 in the revised version.

RC: *2. The Gribov ambiguity has been analyzed in the QRF context by 1809.05093, 2405.13884, and 2410.11029. The presence of inequivalent QRFs is akin to having different superselection sectors, which is analogous to having non-overlapping gauge fixings. The authors should include this in a more extended discussion.*

AR: We thank the referee for pointing us to these references and added them in the footnote addressing the Gribov problem on p. 4, which now reads: “The Gribov obstruction [1] shows that no single such prescription can cover all of the orbits. Note that it has also been explored in the context of QRFs in [2, 3, 4].” Moreover, while we agree with the referee that the connection between QRFs and different superselection sectors is a topic of current interest, we believe it is beyond the scope of the present paper and are looking forward to studying this more in the context of future work.

RC: *3. In the discussion of a symmetry group G , the authors claim that this induces a preferred basis for models. On what grounds is this basis preferred? It may be natural mathematically but I believe the statement has to be qualified as it is as good as any other basis as the authors remark in the remainder of the relevant paragraph.*

AR: We agree with the referee that it is as good as any other basis, as we already point out on p. 6: “In the present paper, we do not explicitly model such other bases. However, in the translation group example, described in

the next Section, where the preferred basis is the position basis, one could straightforwardly express the states and decompose observables in these other bases as well.” Since we are mainly interested in the transformation properties under the action of the symmetry group, it is a particularly illustrative basis for our purposes, though. Nevertheless, and also in response to the first reviewer’s comments, we modified the statement on p. 6 slightly to: *“But of course one can express states and observables in an arbitrary basis. [Footnote: Here, we set aside subtleties that may arise in the context of field theoretic models and in particular in the context of spacetimes in superposition that will be discussed in Sec. V.] In particular, in the translation group example described in the next Section, the preferred basis is the position basis but one could straightforwardly express the states and observables in other bases as well.”*

RC: *4. It is important to note that frame dependence is not a quantum phenomenon. The authors point to superposition and entanglement as being frame-dependent but those are only features of quantum theory which are frame-dependent. This would be good to emphasize at the end of Section II.*

AR: We fully agree with the referee that frame dependence is not a quantum phenomenon. In fact, we explicitly point to this fact in this particular paragraph at the end of section II: “In a sense, however, this extended symmetry is already present at the classical level in the free choice of representational convention. It is at the quantum level, however, that this becomes particularly relevant: for it implies the frame-dependence of properties such as superposition and entanglement.” The fact that superposition and entanglement are frame dependent is a purely quantum phenomenon since superposition and entanglement are.

RC: *5. It might be important to point out that since the authors employ different group elements in different orbits, this leads to the possibility (as in the Gribov problem) to the degeneration of the constraint group. In other words, the theory could break up into different groups $H, G, K, ..$ which gives the model space the complexion of an orbifold.*

AR: We agree with the referee that the existence of stabiliser groups on a subset of the orbits – which we already highlight on p. 12 – changes the topology of the quotient space, turning it into an orbifold (i.e. a stratified manifold). We have included a corresponding footnote highlighting this fact on p. 12 in the main text: “In fact, this means that the space of models can be described as an orbifold (cf. [4]).”

RC: *6. In the second-to-last paragraph of the tenth page, the authors refer to an example of a tripartite system. This was worked out in their Ref. [10] but is not referenced there.*

AR: We agree that this is a very relevant reference at this point and have amended the text accordingly on p. 10.

RC: *7. In Section IV, the authors refer to locally compact groups but then mention quantum diffeomorphisms at the end. Do these constitute a locally compact group?*

AR: Indeed, diffeomorphisms are not a locally compact group – we have highlighted this now on p. 12.

RC: *8. In the second to last paragraph of Sec. IV, the authors discuss the possibility of using coherent states with respect to the group of reference frame orientations, which may not be orthogonal. The authors claim that such discussions have not appeared in the context of QRF they considered and only cite one of the author’s works. Yet such coherent state systems were introduced in this context for temporal reference frames in 1712.00081, and further studied in 1912.00033 and 2007.00580, after which Ref. [14] appeared, which is an extension to unimodular Lie groups built on these results. These works discuss changes of QRFs that are equivalent to what the authors discuss, so it is not clear what the authors mean when they claim “This extension to coherent state systems has not been implemented in the QRF approach of [2,5,8,11,17,18]”. The authors should clarify this point and cite past works where appropriate.*

AR: We are precisely stating that, while the perspective-neutral approach, to which the referee's cited papers can be attributed, *can* capture coherent states, the perspective-dependent approach, developed in the references cited in this paragraph of our manuscript, does not. We thank the referee for pointing out that non-ideal QRFs have been treated in specific contexts in these earlier works and have now included the additional suggested references on p. 12. However, we would like to stress that the equivalence between the perspective-neutral and perspective-dependent approach precisely only holds for ideal frames and not when using coherent states to model the QRF orientations. Thus, we maintain our statement that this extension has not been implemented in the perspective-dependent approach to QRFs.

RC: *9. In the main example of the manuscript, the authors consider a superposition of semiclassical spacetimes. This is a general point of frustration in the literature, which becomes a particular point for the authors, but the description of such objects is still lacking. What does it mean to have a linear Hilbert space for quantum gravity? One can imagine a state with support on multiple backgrounds with overlaps computed using a path integral for instance. However, the description provided here is vague and would be improved if made quantitative.*

AR: We fully agree that the description would be significantly improved if made quantitative and state so at several instances of the manuscript [p. 2, 13, 24]. However, this has been a challenging open problem in the literature for decades and is beyond the scope of the present paper.

RC: *10. The references on the crossed product in quantum gravitational algebras and their relation to QRFs are incomplete. The following references are relevant: 2112.12828, 2206.10780, 2306.07323, 2312.16678, 2405.13884, 2407.01695, 2410.11029, 2207.06704, 2302.14747, 2304.11845, and 2303.02837.*

AR: We thank the referee for providing a more complete list of references to this topic. We included them at the relevant position on p. 25 in the manuscript.

AR: We do want to thank the referee once again for assessing our work and our response to the previous round of comments. We hope this reply addresses and resolves any remaining points.

References

- [1] V. Gribov, Quantization of non-abelian gauge theories, *Nuclear Physics B* **139**, 1–19 (1978).
- [2] A. Vanrietvelde, P. A. Höhn, and F. Giacomini, Switching quantum reference frames in the N-body problem and the absence of global relational perspectives, *Quantum* **7**, 1088 (2023).
- [3] S. Ali Ahmad, W. Chemissany, M. S. Klinger, and R. G. Leigh, Quantum reference frames from top-down crossed products, *Phys. Rev. D* **110**, 065003 (2024), arXiv:2405.13884 [hep-th] .
- [4] S. Ali Ahmad, W. Chemissany, M. S. Klinger, and R. G. Leigh, Relational Quantum Geometry (2024), arXiv:2410.11029 [hep-th] .